# EFFICIENT HUMAN-AI COORDINATION VIA PREPARATORY LANGUAGE-BASED CONVENTION

## ABSTRACT

Developing intelligent agents capable of seamless coordination with humans is a critical step towards achieving artificial general intelligence. Existing methods for human-AI coordination typically train an agent to coordinate with a diverse set of policies or with human models fitted from real human data. However, the massively diverse styles of human behavior present obstacles for AI systems with constrained capacity, while high quality human data may not be readily available in real-world scenarios. In this study, we observe that prior to coordination, humans engage in communication to establish *conventions* that specify individual roles and actions, making their coordination proceed in an orderly manner. Building upon this observation, we propose employing the large language model (LLM) to develop an action plan (or equivalently, a convention) that effectively guides both human and AI. By inputting task requirements, human preferences[1], the number of agents, and other pertinent information into the LLM, it can generate a comprehensive convention that facilitates a clear understanding of tasks and responsibilities for all parties involved. Furthermore, we demonstrate that decomposing the convention formulation problem into sub-problems with *multiple* new sessions being sequentially employed and human feedback, will yield a more efficient coordination convention. Experimental evaluations conducted in the *Overcooked-AI* environment, utilizing a human proxy model, highlight the superior performance of our proposed method compared to existing learning-based approaches. When coordinating with real humans, our method achieves better alignment with human preferences and an average performance improvement of 15% compared to the state-of-the-art.

## 1 INTRODUCTION

Training intelligent agents that can effectively coordinate with humans Carroll et al. (2019) is crucial for enhancing productivity in human society and represents one of the most significant challenges in the pursuit of artificial general intelligence (Goertzel & Pennachin, 2007; Endsley, 2023). Previous approaches to human-AI coordination can be broadly classified into three main directions (Hu & Sadigh, 2023). The first direction involves directly fitting human behaviors or intentions using real human data (Hu et al., 2022; Parekh & Losey, 2023). The second direction focuses on designing algorithms or reward functions inspired by cognitive science to generate human-like policies (Hu et al., 2021; Cui et al., 2021; Laidlaw & Dragan, 2022; Yu et al., 2023). The third direction, known as Population-Based Training (PBT) (Jaderberg et al., 2017), entails constructing a diverse pool of teammates and training a common best response policy (Heinrich et al., 2015; Zhao et al., 2023a). In recent years, researchers have developed various algorithms around these directions, leading to remarkable advancements in human-AI coordination across various domains, including industrial assembly lines (Nourmohammadi et al., 2022), healthcare (Gleichauf et al., 2022), and video games (Siu et al., 2021), etc.

However, existing methods in these directions have certain limitations. In real-world scenarios, obtaining high-quality human data is not always easily accessible, which hampers the feasibility of

---

[1]Human preference here refers to how humans lean towards collaborating to accomplish tasks and the specific roles they undertake. For example, on Overcooked-AI, human-preference can be that the human prefers to make onion soup instead of tomato soup.

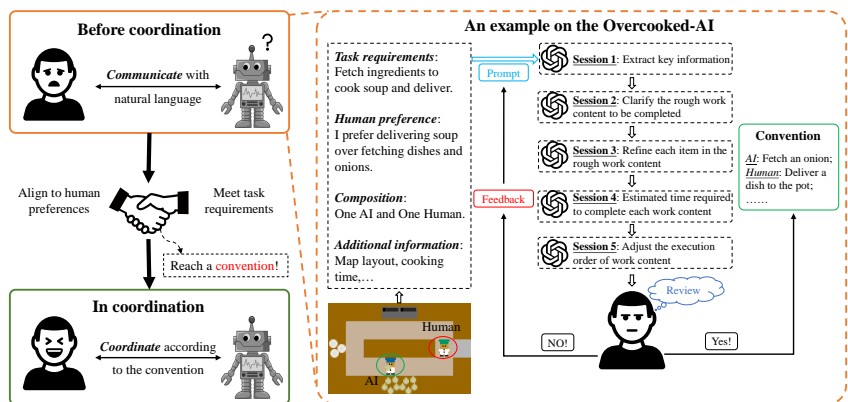

Figure 1: Overview of our proposed HAPLAN on the Overcooked-AI.

fitting human models (Strouse et al., 2021). Moreover, the mechanisms underlying human behaviors are complex (Thatcher & John, 2021). Although incorporating limited representative factors such as attention and irrationality into algorithm design can produce policies that resemble humans more than vanilla reinforcement learning (Sutton & Barto, 2018), it still falls considerably short of capturing true human policies (Laidlaw & Dragan, 2022). Benefiting from the widely proven instance generation ability of PBT (Jaderberg et al., 2017), many attempts been successfully applied (Strouse et al., 2021; Zhao et al., 2023a; Yu et al., 2023; Xue et al., 2022; Charakorn et al., 2022). Despite of this, the challenges associated with PBT methods include: Firstly, maintaining a diverse pool of teammates is not a trivial task, as the policies within the pool need to exhibit sufficient diversity and cover a wide range of real human policies (Yu et al., 2023). Secondly, the policies trained through PBT only have experience in coordinating with teammates from the pool, resulting in poor generalization when encountering unseen teammates (Zhang et al., 2023c). Note that we humans, before coordination, often reach a *convention* (Shih et al., 2020; Gordon, 2023)[2] through communication to determine each individual's task and how to coordinate with each other. Therefore, a natural question arises: *Can we enhance human-AI coordination via human-AI communication?* This is not trivial since humans excel at communicating using natural language, whereas AIs are not proficient in it (Weng, 2020). None of existing methods of the above three directions have the potential to deal with this issue, so we need to find a new way for humans and AIs to discuss and coordinate.

Recently, Large Language Models (LLMs) (Zhao et al., 2023b) have demonstrated impressive capabilities of natural language processing and task planning across various domains, such as robot control (Huang et al., 2023), reasoning (Qiao et al., 2023), and long conversation understanding (Lee et al., 2023), indicating their potential as bridges for human-AI communication and coordination. Hence, this work considers whether we can effectively apply LLMs to facilitate human-AI coordination. A naive approach is to input task requirements, human preferences, the number of AI agents, and other relevant information into an LLM before human-AI coordination, and request it to devise a convention, based on which humans and AIs will coordinate. We conducted experiments on this idea (please see Section 5 for more details), but found it not work well, especially for hard scenarios. We suspect that although the reasoning and planning abilities of the current LLMs are greatly enhanced by emergent techniques such as Chain-of-Thought (Wei et al., 2022), Least-to-Most (Zhou et al., 2023), it still suffers when dealing with challenging problems. When faced with complex tasks or lengthy conversation histories, it struggles to handle them well, and will generate inefficient conventions due to model hallucination (Zhang et al., 2023b).

To tackle the above issues, we propose efficient **H**uman-**AI** coordination via **P**reparatory **LA**nguage-based conventio**N** (**HAPLAN**), a novel framework for human-AI coordination via language-based conventions. When meeting a new task, HAPLAN will first decompose the formulation of a convention into several sub-problems and allocate each of them to a new session separately. Sequentially, the solution of one sub-problem will be the input of another session, which as a result develops a convention specifying roles and assignments for all involved parties. Additionally, to mitigate mistakes made by the LLM, HAPLAN will ask humans to inspect the proposed convention. If any issues are identified, humans will provide feedback to the LLM and require it to reformulate a convention,

---

[2]We use "convention" to refer to the action plan for humans and AIs.

realizing explicit human-ai bidirectional value alignment (Yuan et al., 2022). Figure 1 illustrates an example of our proposed HAPLAN. Comparing with the above naive approach, HAPLAN has at least two advantages: First, each session only needs to handle a simple sub-problem, reducing the probability of making mistakes. Second, by leveraging human feedback, we can refine the planning results from the LLM, thereby improving the effectiveness of human-AI coordination. To evaluate the effectiveness of our approach, we conduct extensive experiments with human proxy models on five maps from Overcooked-AI (Carroll et al., 2019), a generally used benchmark for human-AI coordination. The results show that HAPLAN significantly outperforms existing approaches. Furthermore, by inviting real human players, we observe that HAPLAN achieves higher performance and a better alignment with different humans' preferences. Surprisingly, we find that the idea of decomposing a problem into several sub-problems and assigning them to different sessions can benefit other domains besides human-AI coordination. Experiments on benchmarks of symbolic manipulation, compositional reasoning and math reasoning demonstrate the generality of our idea.

## 2 RELATED WORK

**Human-AI Coordination** Existing works on human-AI coordination can be broadly categorized into three main directions (Hu & Sadigh, 2023). The first direction is to model human behaviors and biases from real human data (Carroll et al., 2019; Hu et al., 2022). However, high-quality human data may not be readily available before human-AI coordination in real-world scenarios. In this work, we consider the setting where there are *no* data of human-AI, human-human, AI-AI coordination. The second direction focuses on designing algorithms or reward functions inspired by cognitive science to generate human-like policies (Hu et al., 2021; Cui et al., 2021; Laidlaw & Dragan, 2022; Yu et al., 2023). Nevertheless, human behaviors are determined by various factors and complex mechanisms (Thatcher & John, 2021). Although taking things like irrationality (Laidlaw & Dragan, 2022), risk sensitivity (Qiu et al., 2021) into consideration will generate policies that resemble humans more than vanilla RL, it is still difficult to fully capture the characteristics of human behaviors. Different from them, our method will ask humans to give their preferences to the LLM and review the proposed conventions, ensuring an effective human-AI coordination with a better alignment to human biases. The third direction, known as Population-Based Training (PBT), entails constructing a diverse pool of teammates and training a common best response policy (Heinrich et al., 2015; Zhao et al., 2023a). Maintaining the diversity of teammates pool under the requirement of covering human policies is not trivial. Moreover, there is no guarantee on the generalizability of the trained policy to unseen humans since it has only coordinated with teammates from the pool.

**Reasoning and Task Planning via LLMs** Recently, Large Language Models (LLMs) has emerged as powerful tools in different domains (Zhao et al., 2023b). Reasoning is an essential ability for complex problem-solving (Qiao et al., 2023). To improve the reasoning ability of LLMs, Wei et al. (2022) proposes Chain-of-Thought to encourage LLMs to explain their reasoning process. We have also taken this idea when designing prompts for multiple sessions (please see Appendix G for more details). However, when solving problems harder than the exemplars shown in the prompts, Chain-of-Thought tends to perform poorly. Zhou et al. (2023) proposes to break down a complex problem into a series of simpler sub-problems and then solve them in sequence, named as Least-to-Most. It requires the LLM to solve all the sub-problems in one session, while our method assigns each sub-problem to a new session separately. There are also some works considering utilizing LLMs to do task planning as our work does, such as Raman et al. (2022) and Huang et al. (2022). But none of them considers human-AI coordination tasks. Some works also try to enable multi-agent coordination with LLMs. Li et al. (2023) proposes to use an LLM to generate and assign sub-goals for AI-AI coordination. Zhang et al. (2023a) considers human-AI coordination, and proposes to integrate an LLM into the field of AI, serving to anticipate humans' forthcoming decisions. We instead use an LLM to make conventions. For users who are interested in LLMs, we recommend to refer to up-to-update surveys such as Zhao et al. (2023b), Wang et al. (2023) and Xi et al. (2023).

## 3 PRELIMINARIES

**Two-Player Human-AI Cooperative Game** In this work, we focus on two-player human-AI coordination, which can be modeled as a two-player Markov decision process extend form markov

games Littman (1994), denoted by $\mathcal{M} = \langle I, \mu_0, \mathcal{S}, \mathcal{A}, \mathcal{P}, R, \gamma \rangle$. Here, $I = \{A, H\}$ is the set of players, where we use $A$ to denote the AI and $H$ to denote the human. $\mu_0$ is the initial state distribution; $\mathcal{S}$ is the state space; $\mathcal{A} = \mathcal{A}^{(A)} \times \mathcal{A}^{(H)}$ is the action space; $\mathcal{P} : \mathcal{S} \times \mathcal{A} \to \Delta_{\mathcal{S}}$ is the transition function[3]; $R : \mathcal{S} \times \mathcal{A} \to \mathbb{R}$ is a global reward function shared by the human and the AI; $\gamma \in [0, 1)$ is the discount factor. Let $\pi_A : \mathcal{S} \to \Delta_{\mathcal{A}^{(A)}}$ be the AI's policy and $\pi_H : \mathcal{S} \to \Delta_{\mathcal{A}^{(H)}}$ be the human's policy. We can define the expected discounted return as $J(\pi_A, \pi_H) = \mathbb{E}\left[\sum_{t=0}^{\infty} \gamma^t R(s_t, a_t^{(A)}, a_t^{(H)})\right]$, where $s_0 \sim \mu_0, a_t^{(A)} \sim \pi_A(\cdot|s_t), a_t^{(H)} \sim \pi_H(\cdot|s_t), s_{t+1} \sim \mathcal{P}(\cdot|s_t, \{a_t^{(A)}, a_t^{(H)}\})$. The goal is to specify $\pi_A$ and $\pi_H$ to achieve the highest $J(\pi_A, \pi_H)$. Here, by saying "specify", we mean to develop a convention with human's preferences being satisfied for both human and AI.

**Convention-based Human-AI Coordination** Inspired by human-human coordination, we consider making conventions for human-AI coordination. Specifically, a convention is a detailed action plan that assigns roles, tasks and other coordination details for both human and AI. Since LLMs are proficient in manipulating natural language, which is convenient for describing high-level plans, but not low-level control instructions, we use an LLM for human-AI coordination in a *hierarchical* manner. That is, the convention proposed by the LLM describes high-level plans. To translate the convention into actions, human can harness his/her ability of natural language understanding, whereas AI relies on pre-trained low-level policies.

## 4 METHOD

This section describes details of our proposed method, HAPLAN. We will begin with an introduction to the prompt designation, the manipulation of multiple sessions and the whole pipeline. Then, we proceed to explain how to train the low-level policies.

### 4.1 TASK PLANNING WITH MULTIPLE SESSIONS

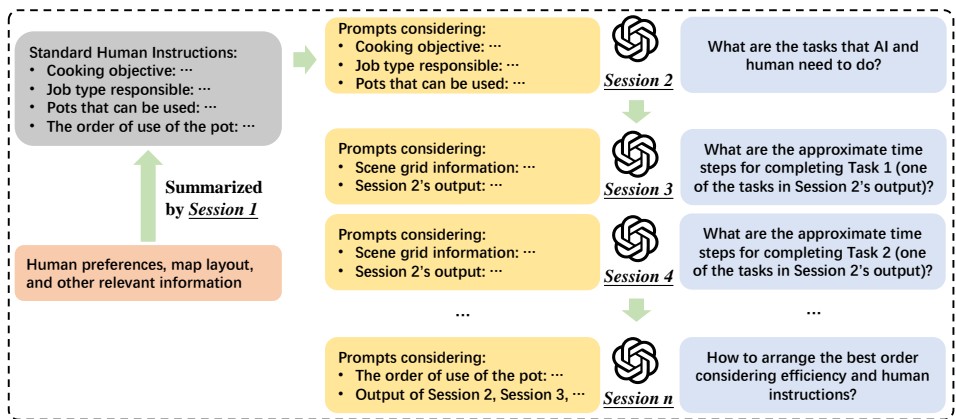

Figure 2: An example of task planning via multiple sessions. The boxes in blue denotes what questions we actually expect LLM sessions to answer, while the boxes in yellow denotes what are contained in the prompts.

**Planner based on Multiple Sessions** Extensive empirical results have shown that when dealing with a complex problem or a long conversation history, the LLM may struggle to effectively cope, leading to the generation of misleading contents (Zhang et al., 2023b). Although methods such as Chain-of-Thought (Wei et al., 2022) and Least-to-Most (Zhou et al., 2023) have greatly improved the reasoning capabilities of the LLM, our experiments in Section 5 have revealed that it still faces challenges in addressing more difficult human-AI coordination tasks. To tackle this issue, we propose employing multiple new *sessions* to jointly develop conventions. Specifically, we decompose a complex problem into multiple sub-problems and assign them sequentially to a new session. By doing so, in each session, the LLM only needs to a much simpler sub-problem and shorter prompt,

---

[3]We use $\Delta_X$ to denote the set of probabilities over $X$.

thus alleviating the issue of model hallucination (Zhang et al., 2023b). The solution provided by one session serves as part of the prompt for the subsequent session. Similar to role play (Shanahan et al., 2023), we implement reasoning via multiple sessions by starting new sessions on ChatGPT (OpenAI, 2022), each with a different prompt. A typical decomposition is shown in Figure 2. Figure 3 illustrates a convention developed by the multiple sessions on the Overcooked-AI environment.

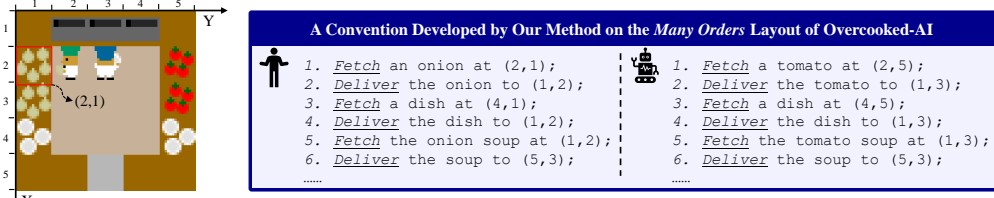

Figure 3: An example of conventions on Overcooked-AI. **Left**: Layout of the *Many Orders* map; **Right**: A convention for human and AI, where the left part is action plans for human and the right part is action plans for AI. $(x, y)$ in the plans denotes the region in the layout whose coordinate on the X-axis is $x$ and coordinate on the Y-axis is $y$.

**Re-plan from Human Feedback** To ensure that the generated convention is both efficient and aligned with human preferences, we incorporate a human validation process. That is, human will review the generated content and provide feedback on any inappropriate aspects. Modifications suggested by human will then be used as part of the prompts for the *first* session to re-plan the convention. We also provide an example of re-planning due to human feedback in Appendix G. Note that both the standard convention establishment and re-planning from human feedback are completed to determine the eventual convention before the episode starts. Allowing human feedback within the episode deserves future research.

## 4.2 EXECUTION WITH PRE-TRAINED SKILLS

The convention contains temporally extended high-level instructions in natural language, which has to be translated into low-level actions executable for AI. To do so, we have pre-trained several skills, similar to SayCan (Ichter et al., 2022). Taking the Overcooked-AI (Carroll et al., 2019) environment as an example, we use the following two skills, visualed in Appendix C:

- `Fetch` This skill empowers the AI to take something at some place. Generally, when there is a need to fetch something, denoted as `A`, at some place, denoted as `B`, we require the LLMs to output a sentence like "`Fetch A at B`".
- `Deliver` This skill enable the AI to take something to some place. When there is a requirement to take something, denoted as `A`, to some place, denoted as `B`, the LLMs will output a plan like "`Deliver A to B`".

We train these two skills using behavioral cloning (Pomerleau, 1991) from human demonstrations. To increase the generalizabilities of the learned skills, we have AI adopt a random policy when collecting human demonstrations. For more implementation details, please refer to appendix C. Depending on the environments, we can also learn more different skills, so our approach is scalable.

## 5 EXPERIMENTS

To validate whether our proposed approach HAPLAN can indeed leverage the advantages of Large Language Models (LLMs) to enhance Human-AI coordination to a new level, we choose Overcooked-AI (Carroll et al., 2019), a test environment commonly used in previous research on human-AI coordination, for empirical experiments. In this environment, there exist two players required to coordinate to complete several order tasks, with each order including a series of steps: fetching ingredients and placing them in the pot, cooking the soup, taking the dishes to scoop the soup and delivering the soup. Specifically, we select five layouts from the environment, and conduct test experiments with both human proxy models and real humans. More details about the Overcooked-AI environment and the selected layouts can be found in Appendix A.1. For baselines, we compare our approach with several popular human-AI coordination algorithms, respec-

Table 1: Experimental results on Overcooked-AI environment of HAPLAN and baselines when coordinating with human proxy policies. The best values have been **bolded**.

| Layout | Partner | FCP | MEP | HSP | HAPLAN |
|---|---|---|---|---|---|
| Counter Circle | Onion Placement | 104.38±9.66 | 133.75±20.27 | 135.38±15.19 | **140.00±26.92** |
| | Delivery | 86.88±9.49 | 83.12±7.26 | 96.25±7.81 | **103.75±10.53** |
| Asymmetric Advantages | Onion Placement & Delivery (Pot1) | 233.13±17.75 | 256.25±18.66 | **282.88±17.03** | 260.63±18.36 |
| | Delivery (Pot2) | 215.00±16.58 | 250.00±19.36 | 258.13±21.71 | **268.00±9.79** |
| Soup Coordination | Onion Placement & Delivery | 199.38±6.09 | 105.00±32.78 | 198.75±4.84 | **219.38±3.47** |
| | Tomato Place & Delivery | 44.38±29.04 | 192.50±9.68 | 128.12±30.76 | **220.63±3.47** |
| Distant Tomato | Tomato Placement | 38.75±30.79 | 27.50±27.27 | 148.75±68.36 | **210.00±15.00** |
| | Tomato Place & Delivery | 175.62±24.35 | 180.00±22.36 | 198.12±37.20 | **251.25±23.41** |
| Many Orders | Tomato Placement | 140.62±32.59 | 170.00±33.91 | 248.75±29.55 | **256.36±35.99** |
| | Delivery | 194.38±12.48 | 175.63±35.61 | 208.13±25.42 | **241.21±12.97** |

tively: Fictitious Co-Play (FCP) (Heinrich et al., 2015), Maximum Entropy Population-based training (MEP) (Zhao et al., 2023a) and Hidden-utility Self-Play (HSP) (Yu et al., 2023). In Appendix B, we introduce the details of these methods.

With the experiments, we aim to answer the following questions: 1) Can our approach obtain better human-AI coordination performance than the existing traditional methods no matter when faced with human proxy models or real humans? (See Section 5.1) 2) Why can the inclusion of LLMs enhance the human-AI coordination performance and what does it bring about? (See Section 5.2) 3) Does utilizing multiple sessions enhance the reasoning capability of LLMs? (See Section 5.3) 4) How does our approach perform on other reasoning benchmarks? (See Section 5.4)

## 5.1 LLMs Enhance Coordination Performance

In this section, we conduct experiments to validate whether the inclusion of LLMs indeed benefits the human-AI coordination. We first introduce the experiments of coordination with human proxy models, and later provide details and results related to experiments with real humans.

### 5.1.1 Coordinating with Human Proxy Models

Firstly, we want to test the ability of our approach to coordinate with partners of different coordination patterns. To serve this purpose, we adopt the scripted policies in HSP (Yu et al., 2023) as the testing partners, which have strong preferences in coordination patterns. To achieve good coordination with these scripted policies, the AI agent must recognize the partner's preference and adapt to it effectively. This poses great challenges to traditional methods, as they do not have an explicit process for knowing about the partner. The experimental results are presented in Table 1.

For fair comparison, the training steps and the pool sizes of FCP, MEP and HSP are all set the same, while our proposed approach HAPLAN is based on LLMs without the need of training one extra coordination policy. The results in Table 1 demonstrates that HAPLAN achieves the highest scores across almost all scenarios. For example, in the layout of *Distant Tomato*, when coordinating with the partner that prefers to place tomatoes in the pot, HAPLAN obtains score several times higher than FCP and MEP, as well as achieves a performance improvement of over 40% compared to HSP. This indicates that methods like FCP and MEP, which train on a pool of partners, struggle to capture specific partner behavior preferences during testing, resulting in a lack of adaptive coordination. HSP is relatively better than them as it explicitly models the human biases. However, our approach still achieves superior performance to HSP, indicating that including LLMs allows better adaptation to various types of partners for improved coordination.

### 5.1.2 Coordinating with Real Humans

In addition to the scripted proxies, we also conducted experiments with real human participants to evaluate the effectiveness of different methods in real human-AI coordination scenarios. Compared to the scripted agents, human players are more flexible and dynamic, making coordination with real human more challenging. In specific, we involve a total of 20 volunteers in the experiment, each of whom had limited prior experience with the *Overcooked-AI* game before. To test a method on one specific layout, we allow the human player to have three rounds of coordination with the AI agent,

Table 2: Experimental results on Overcooked-AI environment of HAPLAN and baselines when coordinating with real humans. The best values in each round of coordination have been **bolded**.

| | | Counter Circle | Asymmetric Advantages | Soup Coordination | Distant Tomato | Many Orders |
|---|---|---|---|---|---|---|
| **First Round** | FCP | 120.00±14.14 | 335.00±21.79 | 190.00±22.36 | 315.00±21.79 | 335.00±32.78 |
| | MEP | **140.00±24.49** | 340.00±14.14 | 180.00±14.14 | 310.00±22.36 | 320.00±28.28 |
| | HSP | **140.00±14.14** | **350.00±33.16** | 185.00±16.58 | **330.00±22.36** | 340.00±31.62 |
| | HAPLAN | 135.00±8.66 | 345.00±16.58 | **195.00±8.66** | 325.00±29.58 | **350.00±53.85** |
| **Second Round** | FCP | 135.00±21.79 | 350.00±17.32 | 190.00±10.00 | 340.00±14.14 | 340.00±24.49 |
| | MEP | 155.00±16.58 | 350.00±22.36 | 185.00±8.66 | 330.00±17.32 | 340.00±28.28 |
| | HSP | 155.00±21.79 | **360.00±14.14** | 195.00±8.66 | 345.00±21.79 | 370.00±22.36 |
| | HAPLAN | **160.00±14.14** | **360.00±24.49** | **205.00±8.66** | **355.00±16.58** | **380.00±50.99** |
| **Third Round** | FCP | 130.00±17.32 | 350.00±22.36 | 200.00±20.00 | 335.00±16.58 | 350.00±22.36 |
| | MEP | 160.00±14.14 | 365.00±16.58 | 195.00±8.66 | 340.00±14.14 | 350.00±36.05 |
| | HSP | 165.00±16.58 | 370.00±22.36 | 200.00±14.14 | 350.00±17.32 | 375.00±25.98 |
| | HAPLAN | **170.00±17.32** | **385.00±21.79** | **215.00±16.58** | **370.00±22.36** | **410.00±51.96** |

allowing us to observe the changes in coordination baselines. Unlike other baseline algorithms, when testing our method, we allow the human partner to engage in natural language communication with the AI agent before the start of each coordination round. The final results are shown in Table 2.

From the experimental results, we can mainly conclude two points: 1) Firstly, under the same number of rounds, our method generally achieves better coordination performance with the human partner. 2) Secondly, our method exhibits a more significant performance improvement through the three rounds of human-AI coordination. In specific, our approach consistently outperforms the baseline algorithms across all layouts after the second round and on some specific layouts our approach attains the best coordination performance right from the first round. For example, on the *Many Orders* layout, HAPLAN achieves the highest score in each round and demonstrates the largest performance improvement across three rounds. This reveals that on one hand the inclusion of LLMs can facilitate the AI agent's coordination with real human partner; on the other hand, LLMs make the AI's behavior more interpretable, helping the humans become familiar with and adapt to the task more quickly. A deeper discussion about why LLMs bring about such gains is provided in Section 5.2.

## 5.2 ANALYSIS OF LLMS IN HUMAN-AI COORDINATION

In fact, the previous traditional methods to some extent separate AI from humans, leaving AI agent an incomprehensible black box for human. The inclusion of LLMs strengthens the interaction between human and AI, allowing both human and AI to understand and benefit each other. In this section, we show the role of LLMs in human-AI coordination, and analyse why LLMs can enhance the coordination performance.

**AI to Human: Explainable AI behaviors** In Figure 4, we let volunteers conduct 5 rounds of tests on the *Asymmetric Advantages* layout, where the results show that our method obtains the fastest score improvement. Besides, we also provide an example to explain the details in this process. This case reveals that when

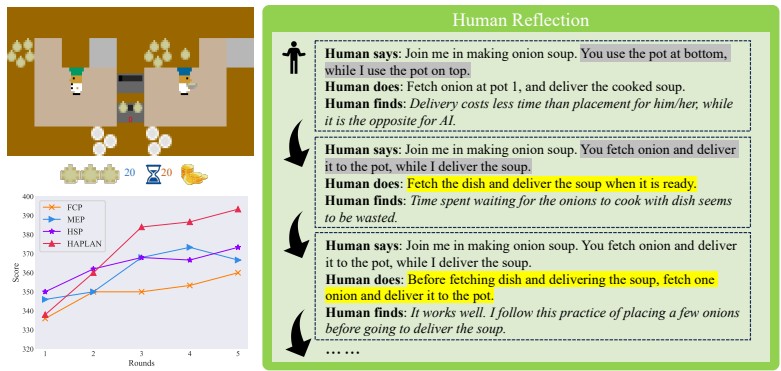

Figure 4: Details of results on the *Asymmetric Advantages* layout.

the AI agent adopts predictable behavioral actions, the human participant can quickly familiarize themselves with the task by trying his/her ideas, and gradually figure out strategies that can be effectively deployed alongside AI. In contrast, for the traditional methods, though the human participant becomes more familiar with the task, it remains challenging to discern how to coordinate with the AI agent since their behavior is difficult to comprehend and unexpected.

**Human to AI: Incorporating human domain knowledge** On the other hand, the utilization of LLMs can help incorporate the human partner's domain knowledge into the human-AI coordination, which can help discover some coordination patterns that are challenging for traditional learning methods to explore. In some complex scenarios, this can significantly contribute to achieving a higher level of human-AI coordination performance. Taking *Many Orders* layout as an example, humans intuitively tend to believe that actively utilizing all three pots is essential for completing the task efficiently. With this insight, as shown in Figure 5(a), our method can achieve exceptionally high scores after the third round, significantly surpassing the highest score of other methods. Thus, from this perspective, the inclusion of LLMs can help incorporate the human's domain knowledge into the coordination, enabling achieving near optimal performance even in some complex scenarios.

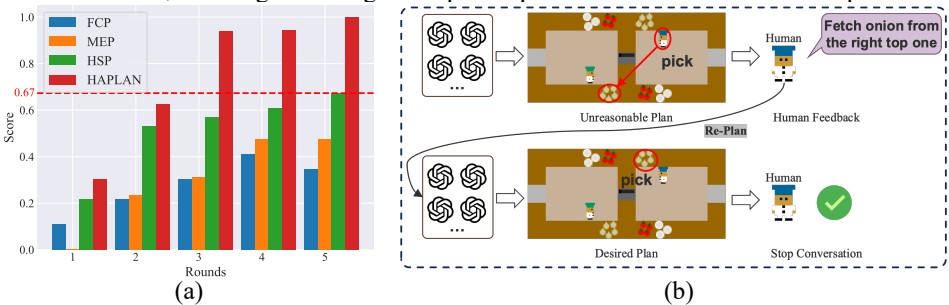

Figure 5: (a) Normalized scores on *Many Orders*. (b) An example of human-AI conversation.

**Human-AI value alignment** Moreover, by introducing LLMs, our method can achieve human-centered human-AI value alignment. That is to say, through multiple rounds of human-AI interactive dialogues, the AI agent can gain a comprehensive understanding of the human partner's thoughts and intentions, ensuring the consistency of the coordination behavior of the entire human-AI coordination system. Such as in the case shown in Figure 5(b), though the human partner expects the AI agent to fetch onions, it is still possible that the AI agent generates unreasonable plan like fetching the onions in the bottom left corner. In such situation, the human partner can continue to correct the AI agent through dialogue, ensuring a desired plan for the AI agent. Similar conclusions can be observed in other layouts. To further provide a quantitative analysis, we present the value alignment results of different methods in the *Many Orders* layout in Figure 6. The results indicate that our method exhibits behavior patterns closest to human value expectation. Totally, our method achieves better human-AI value alignment results, which holds significant value in ensuring the reliability and consistency of the entire human-AI coordination system.

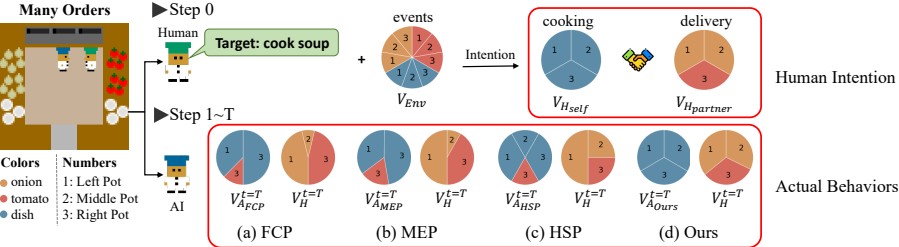

Figure 6: Overview of the human-AI value alignment. Colors denote task types and numbers indicate pot usage, e.g., the red sector of label 1 means placing onions to pot 1, the blue sector of label 2 means delivering the soup in pot 2. $V_{H_{self}}$ and $V_{H_{partner}}$ denote the human's initial intention regarding what they do respectively. Subsequent pie charts show actual event proportions post-trajectory.

## 5.3 ABLATION STUDY FOR UTILIZING MULTIPLE SESSIONS

Our method HAPLAN proposes to use multiple sessions of the LLM to make conventions, where we first decompose the entire problem into several sub-problems, and then utilize separate sessions to solve each sub-problem. To validate whether our practice of utilizing multiple sessions can enhance the reasoning capability, we compare with one baseline called Integrate-LLM (more details refer to Appendix C.2) that only utilizes one single session. This baseline can be seen as an application that combines Chain-of-Thought (CoT) and Least-to-Most (L2M) on Overcooked-AI. A specific point

to note is that, both CoT and L2M also require the problem decomposition, so is Integrate-LLM. Thus, for a fair comparison, we equally decompose the problem into 4 sub-problems and design the same prompts for both Integrate-LLM and our method. Besides, to further validate how the problem decomposition affects the reasoning results, we have additionally included one comparison that decomposes the whole problem into 5 sub-problems, denoted as HAPLAN-5. In Table 3, we present the accuracy of addressing individual sub-problems and, ultimately, deriving valid conventions when using these three methods. From the results we can see that, our method consistently achieves higher accuracy than Integrate-LLM in all sub-problems as well as the final solution, which reveals that our approach of utilizing multiple sessions demonstrates superior reasoning capability on this task. Besides, HAPLAN-5 achieves even higher reasoning accuracy, indicating that appropriate problem decomposition also benefits the final reasoning quality. Note that since HAPLAN-5 performs better, the results in our main experiments are obtained by HAPLAN-5.

Table 3: Reasoning accuracy on the Overcooked-AI environment. "Subprob." denotes sub-problem. The preference of "Placement: 1, Delivery: None" means requiring to place ingredients in pot 1 without delivery requirement. Here we only provide the results for *Placement* task; the complete results can be found in Appendix D.4.

| Preference | Method | Subprob. 0 | Subprob. 1 | Subprob. 2 | Subprob. 3 | Subprob. 4 | Subprob. 5 | Final Solution |
|---|---|---|---|---|---|---|---|---|
| Placement: 1 Delivery: None | Integrate-LLM | / | 40% | 100% | 90% | 50% | / | 0% |
| | HAPLAN | / | 80% | 100% | 100% | 60% | / | 60% |
| | HAPLAN-5 | 100% | 90% | 100% | 100% | / | 100% | 90% |
| Placement: 2+3 Delivery: None | Integrate-LLM | / | 30% | 100% | 100% | 40% | / | 0% |
| | HAPLAN | / | 80% | 90% | 80% | 70% | / | 60% |
| | HAPLAN-5 | 80% | 80% | 100% | 100% | / | 100% | 80% |
| Placement: 1+2+3 Delivery: None | Integrate-LLM | / | 90% | 90% | 100% | 100% | / | 90% |
| | HAPLAN | / | 100% | 100% | 100% | 100% | / | 100% |
| | HAPLAN-5 | 100% | 100% | 100% | 100% | / | 100% | 100% |

## 5.4 ADDITIONAL RESULTS ON REASONING BENCHMARKS

To validate the generality of our approach, we further conduct evaluation on several popular reasoning benchmarks. Special note that these benchmarks are independent of human-AI collaboration and are solely employed to validate the concept of multiple sessions. More details about these benchmarks and the complete experimental results can be found in Appendix A.2 and D.5 respectively. Here, the results on the *Symbolic Manipulation* benchmark

Table 4: Reasoning accuracy on the *Symbolic Manipulation* benchmark.

| Method | Number of Words | | | | |
|---|---|---|---|---|---|
| | L=4 | L=6 | L=8 | L=10 | L=12 |
| Standard prompting (original paper) | 0 | 0 | 0 | 0 | 0 |
| Chain-of-Thought (original paper) | 84.2 | 69.2 | 50.2 | 39.8 | 31.8 |
| Least-to-Most (original paper) | 94 | 88.4 | 83 | 76.4 | 74 |
| Least-to-Most (GPT-3.5) | 100 | 100 | 85 | 70 | 75 |
| Ours (2 Session) | 100 | 100 | 100 | 95 | 95 |

are presented in Table 4. Besides the results reported in the original paper, we also reproduce the Least-to-Most method using the latest GPT-3.5 model and find that it exhibits some performance improvement compared to the results reported in its original paper. We hypothesize that this improvement comes from the update of GPT-3.5. Despite this, our method that utilizes 2 sessions obtains the best performance in all cases with different numbers of words. Moreover, our method also exhibits minimal performance degradation when the number of words increases, still achieving an accuracy of up to 95% which is more than 20% higher than that of Least-to-Most. The results demonstrate the effectiveness of employing multiple sessions for reasoning tasks in various domains.

## 6 CONCLUSION

We propose HAPLAN, an efficient approach to making preparatory language-based conventions for human-AI coordination. To improve the reasoning abilities of LLMs, we propose to decompose a complex problem into several sub-problems and assign each of them to a new session sequentially. For a more efficient coordination, we propose to incorporate a human validation process to review the developed conventions. Experiments on the Overcooked-AI with human proxy models demonstrate the superiority of our approach compared with baselines. When coordinating real humans, our method also achieves higher performance with a better alignment to human preferences. Furthermore, we find that our idea can also be used to solve general reasoning tasks and show its effectiveness on benchmarks of symbolic manipulation, compositional reasoning and math reasoning. At this stage, HAPLAN also has some limitations, which are elaborated in Appendix F

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

# Appendix

## A ENVIRONMENT DETAILS

### A.1 OVERCOOKED-AI ENVIRONMENT

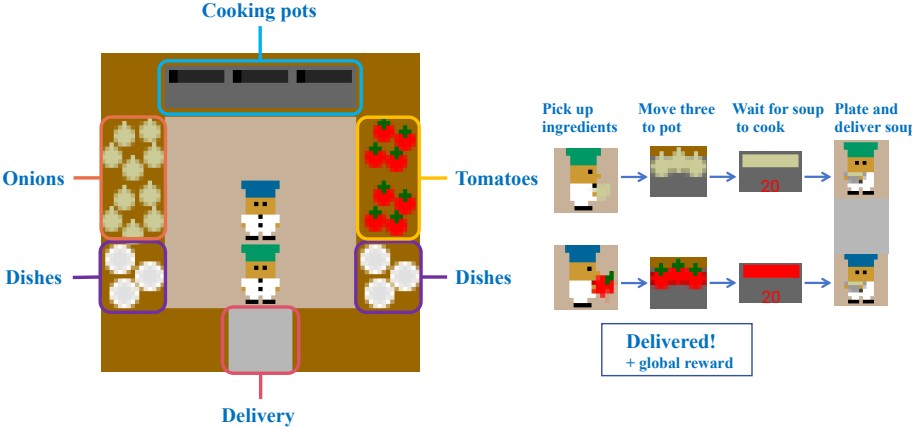

Figure 7: Overcooked-AI (Carroll et al., 2019) environment.

Overcooked-AI (Carroll et al., 2019) is a recent popular benchmark environment for human-AI coordination performance. In this environment, the goal of two players is to complete as many orders as possible within limited time, where each order corresponds to delivering a soup. In specific, each soup requires fetching ingredients (e.g., onions, tomatoes), placing them in the pot, waiting the soup to cook, then picking up the soup and delivering it. The players must coordinate well to complete each step efficiently. Below, we concretely introduce the five layouts used in our experiments.

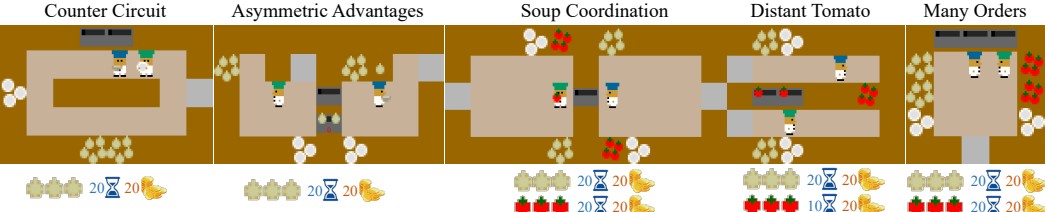

Figure 8: Five layouts of the Overcooked-AI (Carroll et al., 2019) environments used in our experiment. Respectively: *Counter Circle*, *Asymmetric Advantages*, *Soup Coordination*, *Distant Tomato* and *Many Orders*.

**Counter Circle** In the *Counter Circle* layout, two players are placed in the same kitchen. There is a long, narrow counter stretches through the center of the kitchen, necessitating seamless coordination between players to prevent obstruction. Onions, dishes, and pots are positioned at the bottom, left, and top of the kitchen, respectively. Players must employ them appropriately for cooking and delivering the soup. Besides, two pots are located at the top, demanding effective utilization to enhance task completion efficiency.

**Asymmetric Advantages** The *Asymmetric Advantages* layout places two players in two separate kitchens characterized by strong asymmetry. In the left kitchen, it takes more time to fetch the onions as the onions are placed far away from the pot, while the server is close, making delivering convenient. In contrast, in the right kitchen, the situation is reversed: delivery becomes inconvenient, yet fetching the onions is easy. Thus, players can achieve high efficiency through effective coordination.

**Soup Coordination** In the *Soup Coordination* layout, players are also situated into two separate kitchens, yet these two kitchens are essentially symmetrical. Both kitchens contain tomatoes, onions, dishes, and a server, but only the area between the two kitchens features a pot, requiring players to coordinate effectively to complete orders.

**Distant Tomato** In the *Distant Tomato* layout, the two players similarly find them in two separate kitchens. In each kitchen, the onions, dishes, and pots are conveniently close, while the tomatoes are situated at a distance. Furthermore, there are two pots between the two kitchens, requiring both players to coordinate effectively to improve order completion.

**Many Orders** The *Many Orders* layout places players in the same kitchen, with onions and dishes on the left, tomatoes and dishes on the right, and three pots at the top. In this arrangement, players must coordinate effectively to ensure that all three pots are actively used, allowing for efficient soup cooking and delivery.

## A.2   REASONING BENCHMARKS

These reasoning benchmarks are originally introduced in the paper of Least-to-Most (Zhou et al., 2023), including three types of tasks: *Symbolic Manipulation*, *Compositional Generalization* and *Math Reasoning*.

**Symbolic Manipulation** The *Symbolic Manipulation* task typically requires to concatenating the last latter of a series of words. In specific, the input for the LLM is a list of words and the corresponding expected output is the concatenation of the last letters of the words in the list. For example, for inputs "listening, thinking, improve" the corresponding output is "gge", since the last letters of the word list are respectively "g", "g" and "e".

**Compositional Generalization** The *Compositional Generalization* task utilizes SCAN (Lake & Baroni, 2018) as the benchmark, which typically requires mapping natural language command sentences to action sequences. For example, for command "look thrice after jump", the expected action sequence is "JUMP LOOK LOOK LOOK"; for command "run left and walk", the expected action sequence is "TURN_LEFT RUN WALK".

**Math Reasoning** *Math Reasoning* task is aimed to test the reasoning capability of LLMs to solve math world problems in GSM8K (Cobbe et al., 2021) and DROP (Dua et al., 2019). One example question is "Elsa has 5 apples, Anna has 2 more apples than Elsa. How many apples do they have together?"

## B   BASELINE DETAILS

In our main experiments in Section 5.1, we mainly compared our method with three popular human-AI coordination methods, respectively Fictitious Co-Play (FCP) (Heinrich et al., 2015), Maximum Entropy Population-based training (MEP) (Zhao et al., 2023a) and Hidden-utility Self-Play (HSP) (Yu et al., 2023). The introduction of them are as follows:

**FCP** Fictitious Co-Play (FCP) (Heinrich et al., 2015) is a two-stage approach to learn to collaborate with humans without human data. At the first stage, it builds a pool of partners which represent different conventions; while at the second stage, it trains a best-response agent to the obtained diverse partners and their checkpoints.

**MEP** Maximum Entropy Population-based training (MEP) (Zhao et al., 2023a) also follows a two-stage framework, while it proposes to learn a diverse partner population through maximizing one centralized population entropy objective.

**HSP** Hidden-utility Self-Play (HSP) (Yu et al., 2023) explicitly models the human biases as hidden reward functions. On this basis, it augments the policy pool with biased policies and afterwards trains an adaptive policy.

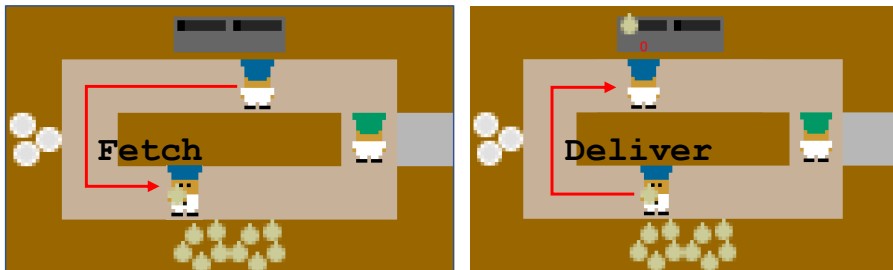

Figure 9: Visualization of the low-level skills, `Fetch` (**left**) and `Deliver` (**right**).

## C   IMPLEMENTATION DETAILS

### C.1   LOW-LEVEL SKILL LEARNING

One significant component of our approach is the building of low-level skills. Only when the low-level skills are reliable, combined with the reasoning of the upper-level LLMs, can our AI agent coordinate effectively with humans. To achieve this goal, we propose a method that involves human players providing a small number of demonstrations, followed by learning lower-level skill policies through imitation learning techniques. This approach can be applied in scenarios where humans can provide a few demonstrations or where there exist a few demonstration segments (only fragmentary data required, **no need for complete coordination trajectory**).

Specifically, in the Overcooked-AI environment, we have learned two types of low-level skills, as shown in Figure 9, by requiring human players to collect a few demonstration trajectories. Each sub-task in this environment can be completed by this two two-level-skills. For example, the Delivery task (pick up the dish, scoop the soup and deliver the soup) in Overcooked-AI can be split into: `Fetch` the dish at `(x1,y1)`, `Deliver` the dish to `(x2,y2)`, `Fetch` the soup at `(x2,y2)`, `Deliver` the soup to `(x3,y3)`. The specific learning process can be divided into the following two steps:

**Data Collection**   To learn these two low-level skills, we firstly collected skill demonstration data by requiring the human participants to play a small number of games. In this process, to ensure the robustness of the learned skill policies when facing situations of different partner states, we had human players interact with one partner of random policy, thereby ensuring that the training data encounters various partner states. In our experiments, the demonstrations for training each skill are around 50 trajectories.

**Imitation Learn**   After collecting the demonstration data, we began to learn the skill policies through behavior cloning method. In details, on one hand, we allow the skill parameters to be part of input into the skill policy. For example, for skill "`Fetch` an onion at `(2,1)`", we will concatenate the coordinate information `(2,1)` with the agent's observation and input them into the skill policy. On the other hand, we applied data augmentation to the demonstration data to further enhance the robustness of the learned skill policies. For example, for demonstration of fetching dish, if the pot is empty, we will augment new trajectory data with pot filled with ingredients. The reason is that whether there are ingredients in the pot does not affect the action of fetching the dish.

### C.2   IMPLEMENTATION OF INTEGRATE-LLM

In the ablation study of utilizing multiple sessions on Overcooked-AI, we have compared our method with one baseline called Integrate-LLM as mentioned in Section 5.3. Due to space constraints, we did not provide a detailed introduction to Integrate-LLM in the main text. Actually, it is implemented by simply replacing the utilization of multiple sessions in our method with using one single session, which means that this session will sequentially address all the sub-problems, and the relevant information of solving the sub-problems earlier will be retained as part of the prompt for obtaining the final reasoning results. Besides, the prompt engineering employed by Integrate-LLMs is also nearly identical to our approach, which is listed in Appendix G.

Table 5: Complete results of reasoning accuracy on the Overcooked-AI environment.

| Rough preference | Specific preference | Method | Subprob. 0 | Subprob. 1 | Subprob. 2 | Subprob. 3 | Subprob. 4 | Subprob. 5 | Final Solution |
|---|---|---|---|---|---|---|---|---|---|
| Only Placement | Placement: 1 Delivery: None | Integrate-LLM | / | 40% | 100% | 90% | 50% | / | 0% |
| | | HAPLAN | / | 80% | 100% | 100% | 60% | / | 60% |
| | | HAPLAN-5 | 100% | 90% | 100% | 100% | / | 100% | 90% |
| | Placement: 2+3 Delivery: None | Integrate-LLM | / | 30% | 100% | 100% | 40% | / | 0% |
| | | HAPLAN | / | 80% | 90% | 80% | 70% | / | 60% |
| | | HAPLAN-5 | 80% | 80% | 100% | 100% | / | 100% | 80% |
| | Placement: 1+2+3 Delivery: None | Integrate-LLM | / | 90% | 90% | 100% | 100% | / | 90% |
| | | HAPLAN | / | 100% | 100% | 100% | 100% | / | 100% |
| | | HAPLAN-5 | 100% | 100% | 100% | 100% | / | 100% | 100% |
| Only Delivery | Placement: None Delivery: 2 | Integrate-LLM | / | 40% | 90% | 80% | 20% | / | 0% |
| | | HAPLAN | / | 70% | 100% | 100% | 100% | / | 70% |
| | | HAPLAN-5 | 70% | 70% | 100% | 100% | / | 100% | 70% |
| | Placement: None Delivery: 1+3 | Integrate-LLM | / | 40% | 100% | 80% | 30% | / | 0% |
| | | HAPLAN | / | 60% | 100% | 100% | 100% | / | 60% |
| | | HAPLAN-5 | 80% | 100% | 100% | 100% | / | 100% | 80% |
| | Placement: None Delivery: 1+2+3 | Integrate-LLM | / | 80% | 100% | 90% | 60% | / | 70% |
| | | HAPLAN | / | 100% | 100% | 100% | 100% | / | 100% |
| | | HAPLAN-5 | 100% | 100% | 100% | 100% | / | 100% | 100% |
| Place & Delivery | Placement: 1 Delivery: 1 | Integrate-LLM | / | 90% | 100% | 100% | 100% | / | 100% |
| | | HAPLAN | / | 100% | 100% | 100% | 100% | / | 100% |
| | | HAPLAN-5 | 100% | 100% | 100% | 100% | / | 100% | 100% |
| | Placement: 1 Delivery: 1+2 | Integrate-LLM | / | 90% | 100% | 90% | 70% | / | 60% |
| | | HAPLAN | / | 100% | 100% | 100% | 100% | / | 100% |
| | | HAPLAN-5 | 100% | 100% | 100% | 100% | / | 100% | 100% |
| | Placement: 2+3 Delivery: 3 | Integrate-LLM | / | 70% | 100% | 100% | 90% | / | 70% |
| | | HAPLAN | / | 80% | 100% | 100% | 100% | / | 80% |
| | | HAPLAN-5 | 100% | 100% | 100% | 100% | / | 100% | 100% |
| | Placement: 1 Delivery: 1+2+3 | Integrate-LLM | / | 90% | 100% | 90% | 90% | / | 90% |
| | | HAPLAN | / | 100% | 100% | 100% | 100% | / | 100% |
| | | HAPLAN-5 | 100% | 100% | 100% | 100% | / | 100% | 100% |

This practice resembles Least-to-Most as they both utilize one single LLM (one session) to sequentially solve multiple sub-problems in order to solve the whole problem. The difference lies in that Least-to-Most directly applies LLM to decompose the problem, while Integrate-LLM employs a pre-defined problem decomposition scheme. Besides, it includes an example of problem solving that aligns with the problem decomposition scheme in the beginning of the prompt, which is similar to the practice of the Chain-of-Thought method. Thus, Integrate-LLM can be seen as an application that combines Chain-of-Thought and Least-to-Most on Overcooked-AI. The comparison with this baseline demonstrates that utilizing only one single session may struggle in some complex scenarios.

# D ADDITIONAL RESULTS

## D.1 ABLATION STUDY ON CONVENTION MECHANISM

To investigate to what extent the convention is actually helping in our approach, we externally include one baseline denoted as HAPLAN w/o convention, which disables developing conventions via human feedback. Instead, it provides a heuristic plan for the agent. The results are shown in Table 6, where we can see that HAPLAN w/o convention suffers from a significant performance decline especially at the final round, indicating the effectiveness of convention.

Table 6: Results of ablation study on convention.

| | | Counter Circle | Asymmetric Advantages | Soup Coordination | Distant Tomato | Many Orders |
|---|---|---|---|---|---|---|
| First Round | HAPLAN w/o convention | 135.00±21.79 | 340.00±14.14 | 195.00±16.58 | 325.00±25.98 | 355.00±45.55 |
| | HAPLAN | 135.00±8.66 | 345.00±16.58 | 195.00±8.66 | 325.00±29.58 | 350.00±53.85 |
| Second Round | HAPLAN w/o convention | 145.00±16.58 | 350.00±10.00 | 205.00±16.58 | 335.00±16.58 | 365.00±29.58 |
| | HAPLAN | 160.00±14.14 | 360.00±24.49 | 205.00±8.66 | 355.00±16.58 | 380.00±50.99 |
| Third Round | HAPLAN w/o convention | 150.00±22.36 | 355.00±8.66 | 205.00±8.66 | 340.00±28.28 | 370.00±33.16 |
| | HAPLAN | 170.00±17.32 | 385.00±21.79 | 215.00±16.58 | 370.00±22.36 | 410.00±51.96 |

## D.2 ANALYSIS OF ORACLE COLLABORATION PERFORMANCE

To delve deeper into the characteristics of the Overcooked-AI environment and evaluate how HAPLAN performs on it, we train a joint policy on Overcooked-AI via MAPPO (Yu et al., 2022) algorithm. The resulting performance is referenced as as "Oracle" since both sides are best responses to each other. The results are listed in Table 7. Notably, despite the human partners not being experts in

Overcooked-AI, HAPLAN can achieve 75%~89% of the oracle performance via planning by LLMs and human feedback.

Table 7: Comparison with the oracle performance on Overcooked-AI environment.

| | Counter Circle | Asymmetric Advantages | Soup Coordination | Distant Tomato | Many Orders |
|---|---|---|---|---|---|
| HAPLAN (First Round) | 135.00±8.66 | 345.00±16.58 | 195.00±8.66 | 325.00±29.58 | 350.00±53.85 |
| HAPLAN (Second Round) | 160.00±14.14 | 360.00±24.49 | 205.00±8.66 | 355.00±16.58 | 380.00±50.99 |
| HAPLAN (Third Round) | 170.00±17.32 | 385.00±21.79 | 215.00±16.58 | 370.00±22.36 | 410.00±51.96 |
| Oracle | 225.19±18.31 | 447.63±8.11 | 240.67±4.13 | 481.51±54.08 | 462.76±12.71 |

## D.3 ANALYSIS OF STRATEGY DIVERSITY IN REAL HUMAN EXPERIMENTS

To investigate the diversity of cooperation strategies during cooperation with different participants in real human experiments, we collect the trajectories in those experiments and conduct visualization analysis for them. In specific, we collect the trajectories of all 16 human participants, those cooperating with FCP/MEP/HSP/HAPLAN methods, in *Many Orders* layout, and conduct T-SNE for visualization. The final visualization result is depicted in Figure 10. Each point, square or triangle in the figure represents one specific trajectory and different colors represent different human participants. The results show that the behavior of human participants is diverse, indicating the ability of HAPLAN to cooperate well with diverse human teammates through establishing conventions with them.

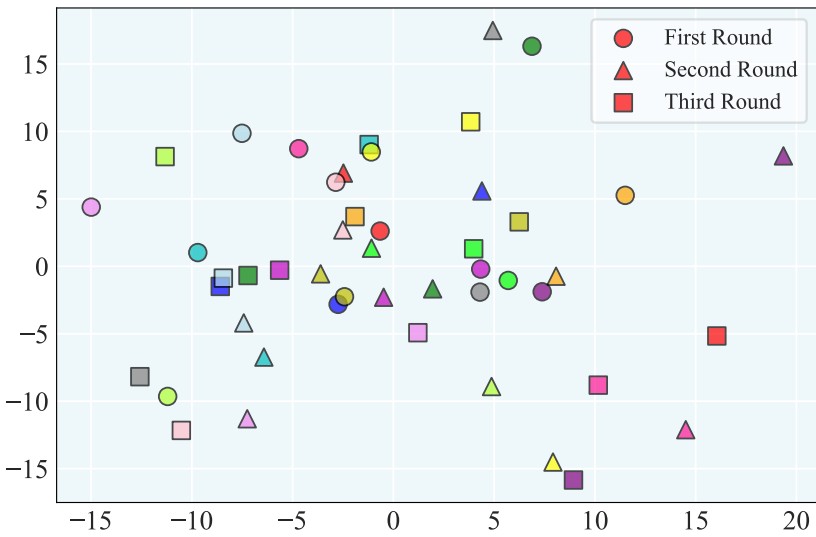

Figure 10: Visualization result of the trajectories in *Many Orders* layout during real human experiments.

## D.4 COMPLETE RESULTS OF REASONING ACCURACY ON OVERCOOKED-AI

To validate whether utilizing multiple sessions of the LLM can enhance the reasoning capability, we conduct ablation to evaluate the reasoning accuracy on Overcooked-AI environment. Specifically, we have multiple tasks where each has certain preferences. We first generate several corresponding commands for each task, and subsequently we evaluate the accuracy of different methods in deducing planning results that fulfill these requirements through reasoning. We have provided some results in Section 5.3, while the complete results are presented in Table 5. It can be observed that our method consistently achieves the best reasoning results in this experiment.

## D.5 COMPLETE RESULTS ON REASONING BENCHMARKS

To validate the generality of our approach, besides the human-AI coordination benchmark Overcooked-AI, we further test our approach on some typical reasoning benchmarks to see whether utilizing multiple sessions can also yield benefits on tasks of other domains. In Section 5.4, we report the experimental results on *Symbolic Manipulation* benchmark, and find that our approach does

Table 8: Reasoning accuracy on the *Compositional Generalization* benchmark.

| LLM-Model | Method | | | | |
|---|---|---|---|---|---|
| | Standard prompting | Chain-of-Thought | Least-to-Most | Ours (2 session) | Ours (3 session) |
| code-davinci-002 (original paper) | 16.7 | 16.2 | **99.7** | / | / |
| GPT-3.5 | 0 | 35 | 25 | 65 | **90** |

enhance the reasoning accuracy of LLMs and obtains superior performance compared to previous methods. The further results on *Compositional Generalization* and *Math Reasoning* are respectively presented in Table 8 and Table 9.

On the *Compositional Generalization* benchmark, we also reproduce the results of Chain-of-Thought and Least-to-Most using the latest GPT-3.5 model. We find that the replicated results differ from the results reported in the paper of Least-of-Most, for which we hypothesize the reason is the update of the GPT-3.5 model. However, when equally employing GPT-3.5, our method of utilizing 2 sessions can be able to obtain accuracy results significantly higher than the baseline algorithms. Moreover, when we add the number of LLMs to 3, our method exhibits even better reasoning performance, achieving accuracy of 90%.

On the *Math Reasoning* benchmark, we find that Standard prompting with the latest GPT-3.5 model achieves significantly higher accuracy than reported in the paper of Least-to-Most, which indicates that now the updated GPT-3.5 model can have been fine-tuned on similar datasets. For this reason, all methods do not show a very significant improvement to Standard prompting on this benchmark. Chain-of-Thought obtains the highest score on Non-Football (DROP), while our method performs slightly better on Football (DROP) and GSM8K.

Table 9: Reasoning accuracy on the *Math Reasoning* benchmarks.

| Method (LLM-Model) | Benchmarks | | |
|---|---|---|---|
| | Non-Football (DROP) | Football (DROP) | GSM8K |
| Standard prompting (original paper) | 58.78 | 62.73 | 17.06 |
| Chain-of-Thought (original paper) | 74.77 | 59.59 | 60.87 |
| Least-to-Most (original paper) | 82.45 | 73.42 | 62.39 |
| Standard prompting (GPT-3.5) | 85.00 | **80.00** | 70.00 |
| Chain-of-Thought (GPT-3.5) | **90.00** | 65.00 | 70.00 |
| Least-to-Most (GPT-3.5) | 85.00 | 70.00 | 65.00 |
| Ours (GPT-3.5) | 85.00 | **80.00** | **75.00** |

# E    MORE DETAILS ABOUT REAL HUMAN EXPERIMENTS

## E.1    RECRUITMENT DETAILS

We recruited human participants from our campus community, specifically by posting a recruitment notice seeking individuals to participate in a Gaming AI test. To mitigate the impact of participants' prior experience, we selected participants who have not played the Overcooked game. We finally recruited 20 volunteers and they fall within the age range of 19 to 22 years old.

Basically, we provide 15 dollars for each participant for compensation. Meanwhile, to encourage participants to conduct testing seriously, we provide extra 10 dollars for the top three participants based on their total score achieved during the gameplay.

## E.2    EXPERIMENT INSTRUCTIONS

To provide participants with a rough background of the experiments, we provide them with experiment instructions as follows:

**Basic operations**    The Overcooked game naturally requires human-AI coordination to achieve a high score. In this game, you will cooperate with an AI agent as a team.

- There are six actions available in the game: up ("W"), down ("S"), left ("A"), right ("D"), no-operation ("F"), and interact ("G"). Each action consumes one time step, and a complete game consists of 400 time steps.

- The relevant objects are onions, tomatoes, plates, and soups.

- When the game starts, you can move (up, down, left, and right) towards the onion and interact with it. Once you interact with the onion, you will acquire it and be carrying it.

- When you are carrying an onion, you can move to a pot and interact. By doing so, you will place the onion you are carrying into the pot.

- Place three onions or tomatoes in a pot and wait for 20 time steps, then the ingredients will transform into a pot of delicious soup.

- Now move to a plate and interact with it to carry the plate. Take the plate and move towards the cooked soup, then interact with it. This action will scoop the soup into the plate.

- The final step is to deliver the vegetable soup to the delivery port, and you will complete an order successfully.

- Upon completing each order, you will receive a reward of 20 points.

**Experimental settings**   In our experiment, you will cooperate with one agent teammate for three rounds on each Overcooked layout. These agents may require interaction with you or not.

- When cooperating with agent that requires interaction, it is necessary to establish a convention with it through the interactive window before each episode. For example, you can say, "I will make onion soup, and the AI will handle food delivery." Based on this convention, the system will generate a detailed plan for you and your AI teammate. You have the option to either accept the plan, request a plan reformulation, or even change your convention.

- When working with other agents, you will directly cooperate with them and strive to achieve high scores.

### E.3   EXPERIMENT SETUP

In real human experiments, we divide the recruited 20 participants into 5 groups. Across each layout, we randomly assigned these 5 groups of participants to different AI methods (total 5 AI methods including HAPLAN w/o convention in Appendix D.1) for testing. In other words, we test each AI method with 4 human participants. Besides, to avoid unfair comparison between different AI methods, we ensured similarities in age and gender distribution among different groups and randomly assigned each group to cooperate with one specific AI method for each layout in experiments.

### E.4   ETHICAL STATEMENT

In terms of the real human experiments, we have adopted effective practice to mitigate potential risks and ethical issues. Actually, the main risks for the human volunteers in our experiments are 1) personal information leakage and 2) time cost. For the former, we only invite the volunteers to participate in the test experiments without requirement for any extra personal information. Also, we will maintain strict confidentiality of the volunteers' identities. For the latter, we improve the interactive interface using the `Flask` framework to enhance the user experience and testing efficiency. Besides, we provide certain material compensation to the volunteers.

For the boarder impact of our approach, as our work has not reached the application stage, it does not have a boarder impact currently. In the future, we will carefully consider the societal impact our method may bring.

## F   LIMITATION AND FUTURE WORK

**Critical Assumptions**   Actually, there exist two critical assumptions for HAPLAN's success that should be specifically emphasized, which are the ability to find high-level actions that can be executed by the low-level controller robustly and also make sense to the user. Previous systems like

SayCan (Ichter et al., 2022) also requires the first assumption, which is kindly feasible in quite a lot scenarios. For the second assumption, despite the skill actions in SayCan are not conveyed to humans, they are also represented in the form of natural language, which implicitly indicates that humans are likely to be able to understand these skill actions. Despite HAPLAN's dependence on these two assumptions, we believe they can be realized in most human-AI coordination scenarios. Moreover, there also exists another assumption that our method can have access to a high-level abstraction of human actions. We make this assumption and assume this part of information can be provided in the background knowledge of the task.

**Application Constraints**   In our approach, HAPLAN requires a detailed conversational coordination phase prior to any actual physical interaction. Thus, it is obviously not suitable for any problem settings. Currently, the application of HAPLAN still faces difficulties in scenarios that demand stricter restrictions on pre-conversation between humans and AI. Moreover, HAPLAN requires finishing conversation before the episode starts, more general approach that allows human feedback within episode deserves future research.

**Automated Problem Decomposition and Convention Evaluation**   In our current implementation, problem decomposition is provided by humans. While our decomposition approach is not coupled with the task and can provide some inspiration for other tasks, specific problems may benefit from more efficient decomposition. One straightforward approach is to employ an additional session to suggest problem decomposition. Moreover, we are having humans review the conventions generated by the LLM, which may encounters challenges in complex problem scenarios. However, it is also possible to consider having the LLM itself review the content it generates and construct AI feedback through other approaches. We plan to leave them for future works.

## G   PROMPT ENGINEERING

**Prompts for the Overcooked-AI.**

==== Prompt for Session 1, to extract key information related to task planning tasks ====

In a collaborative cooking game, you are an AI who needs to play the role of a chef with one human player. Before the game starts, humans will communicate with you, giving you human instructions. Please extract key information from human instructions, including 'Cooking Objectives' and' My Work '.

Making any dish requires using a pot and completing two tasks: 'Fetching vegetables' and 'Delivering food'. Therefore, if humans indicate that you need to make a certain type of food yourself, then you need both 'Fetching vegetables' and 'Delivering food'. Among them, 'Fetching vegetables' refers to placing an uncooked ingredient in a pot for the next step of work. 'Delivering food' refers to the delivery of food from a pot to the delivery port after it has been cooked.

If the cooking objective is tomato soup, then the ingredient to be placed in the pot is tomato; Similarly, if the cooking goal is onion soup, the ingredient to be prepared in the pot is onion.

For Example 1:
The instructions for humans are: Please make tomato soup.
Your answer:
Cooking objectives: tomato soup
AI's jobs:
Fetching vegetables: All pots.
Delivering food: All pots.

For Example 2:
The instructions for humans are: Please make tomato soup, and you are only responsible for preparing tomatoes. Please take the tomatoes from the tomato spot on the right.
Your answer:
Cooking objectives: tomato soup
AI's jobs:

Fetching vegetables: All pots.
Delivering food: Not mentioned.

For Example 3:
The instructions for humans are: Please use the pot on the right to make onion soup.
Your answer:
Cooking objectives: onion soup
AI's jobs:
Fetching vegetables: the pot on the right.
Delivering food: the pot on the right.

For Example 4:
The instructions for humans are: Please use the pot on the left to make onion soup and be responsible for the delivery of the middle pot.
Your answer:
Cooking objectives: onion soup
AI's jobs:
Fetching vegetables: the pot on the left.
Delivering food: the pot on the left + the middle pot.

Now, the instructions for humans are: Please join me in making onion soup. You are only responsible for putting the onion into the pot and do not take onions from the onion dots below. Please provide your answer by giving examples.

---

==== Prompt for Session 2, to clarify the rough work content ====
In a collaborative cooking game, you need to play the role of a chef with one human player. To collaborate better with humans in the game, you need to plan your rough work content with humans before the game starts.
Firstly, you need to clarify the location of the pot mentioned in the key information. Then, based on the job description of AI in the key information, obtain the rough work content of AI. Finally, obtain the rough work content that humans need to complete.

For Example 1:
The pot in the scene: (1,2), (1,3), (1,4)
Key information in human instructions:
Cooking objectives: onion soup
AI's jobs:
(1) Fetching vegetables: the pot on the left.
(2) Delivering food: the pot on the left + the middle pot.

Your answer:
the pot on the left is pot (1,2)
the middle pot is pot (1,3)
So, the rough work contents that AI need to do are:
(1) Fetch onions for pot at (1,2)
(2) Deliver onion soup for pot (1,2)
(3) Deliver onion soup for pot (1,3)
Correspondingly, the rough tasks that humans need to complete are:
(1) Fetch onions for pot at (1,3)
(2) Fetch onions for pot at (1,4)
(3) Deliver onion soup for pot (1,4)

Now, the pot in the scene: (2,3), (1,3), (1,4)
Key information in human instructions:
Cooking objectives: tomato soup
AI's jobs:
(1) Fetching vegetables: the pot below.

(2) Delivering food: other pots.
Please provide your answer by giving examples.

---

==== Prompt for Session 3, to refine the rough work content ====
Please refine the rough work content based on the scenario information.
The rough work content is divided into two categories:
One is to pick up ingredients, which need to be refined to where to take the ingredients from and into which pot.
The second is to deliver food, which needs to be refined to the location from which the plate is taken, then the food is placed in the plate in which pot, and then the food loaded on the plate is sent to which delivery port.
In addition, you also need to consider whether there are restrictions on the items you can use in human instructions.

For Example 1:
Human instructions: Please prepare onions.
The rough work content is: Pick up onions for pot (1,2)
Scenario information is:
Location of Tomatoes: (2,5), (3,5)
Location of Onions: (2,1), (3,1)
Location of the dining plate: (4,1), (4,5)
Location of the delivery port: (5,2)
Your answer:
There are no additional restrictions in the human instructions on where to take onions.
For the first onion position (2,1), its distance from the pot (1,2) is $|2-1|+|1-2| = 1+1 = 2$.
For the second onion position (3,1), its distance from the pot (1,2) is $|3-1|+|1-2| = 2+1 = 3$.
Therefore, I should choose a location (2,1) closer to the pot (1,2) to take the onion.
So, the refined work content is: Take the onion from position (2,1) and place it in the pot (1,2).

For Example 2:
Human instructions: Please use the pot on the right to make tomato soup.
The rough work content is: Deliver tomato soup for pot to (1,3)
Scenario information is:
Location of Tomatoes: (2,5), (3,5)
Location of Onions: (2,1), (3,1)
Location of the dining plates: (4,1), (4,5)
Location of the delivery ports: (5,2)
Your answer:
There are no additional restrictions in the human instructions on where to pick up the plate and which delivery point to deliver it to.
For the first dining plate position (4,1), its distance from the pot (1,3) is $|4-1|+|1-3| = 3+2 = 5$.
For the second dining plate position (4,5), its distance from the pot (1,3) is $|4-1|+|5-3| = 3+2 = 5$.
Therefore, I should choose a location (4,1) closer to the pot (1,3) to take a plate. For the first delivery port (5,2), its distance from the pot (1,3) is $|5-1|+|2-3| = 4+1 = 5$.
Therefore, I should choose the delivery port (5,2) closer to the pot (1,3) to deliver the food.
So, the refined work content is: Take the plate from (4, 1), then take the food from the pot (1, 3), and finally deliver it to the delivery port (5, 2).

For Example 3:
Human instructions: Please prepare onions. You can only take onions from the onion dots below.
The rough work content is: Pick up onions for pot (1,2)
Scenario information is:
Location of Tomatoes: (2,5), (3,5)
Location of Onions: (2,1), (3,1)

Location of the dining plate: (4,1), (4,5)
Location of the delivery port: (5,2)
Your answer:
The human instructions require me to pick onions from the onion dots below.
the onion dots below is (3,1).
So, the refined work content is: Take the onion from position (3,1) and place it in the pot (1,2).

Now, the rough work content is: Deliver tomato soup for pot (1,4)
Scenario information is:
Location of Tomatoes: (2,1), (2,5)
Location of Onions: (3,1), (3,5)
Location of the dining plates: (4,1), (4,5)
Location of the delivery ports: (5,2)
Please provide your answer by giving examples.

---

==== Prompt for Session 4, to calculate the approximate time required to execute each detailed work content ====
There is currently a task in a grid world, please estimate the approximate time required to perform this task.
Among them, each move of a character requires one time step, and interacting with objects in the scene requires one time step.
For two types of work:
(1) Fetching vegetables: the approximate time is six times the time it takes to move the vegetables from their position to the pot position.
(2) Delivering food: the approximate time is from the position of the plate to the position of the pot, to the position of the delivery port, and then to the position of the plate.

For Example 1:
The rough work content is: Pick up onions for pot (1,2)
The refined work content is: Take the onion from position (2,1) and place it in the pot (1,2).
Your answer:
Moving onions from (2,1) to (1,2) requires $|2 - 1| + |1 - 2| = 1 + 1 = 2$ steps.
So, the approximate time is: $2 \times 6 = 12$ steps.

For Example 2:
The rough work content is: Deliver tomato soup for pot (1,3)
The refined work content is: Take the plate from (4, 1), then take the food from the pot (1, 3), and finally deliver it to the delivery port (5, 2).
Your answer:
Moving from (4,1) to (1,3) requires $|4 - 1| + |1 - 3| = 3 + 2 = 5$ steps.
Moving from (1,3) to (5,2) requires $|1 - 5| + |3 - 2| = 4 + 1 = 5$ steps.
Moving from (5,2) to (4,1) requires $|5 - 4| + |2 - 1| = 1 + 1 = 2$ steps.
So, the approximate time is: $5 + 5 + 2 = 12$ steps.

Now, the rough work content is: Deliver tomato soup for pot (1,4)
The refined work content is: Take the plate from (4, 5), then take the food from the pot (1, 4), and finally deliver it to the delivery port (5, 2).
Please provide your answer by giving examples.

---

==== Prompt for Session 5, to adjust the order of work tasks ====
Please adjust the order of execution for the rough work content that needs to be completed.
Please note that there are two rough job descriptions: picking up vegetables and delivering food. However, for the same pot, delivery work can only be carried out 20 time steps after completing the vegetable picking work.

For Example 1:

The rough work contents are:
(1) Fetch onions for pot at (1,2), 12 steps
(2) Deliver onion soup for pot (1,2), 10 steps
(3) Fetch onions for pot at (1,3), 18 steps
Your answer:
Due to the fact that the execution of the work 'Delivery on soup for pot (1,2)' requires 20 time steps after completing the 'Pick up onions for pot (1,2)', in order to fully utilize the waiting time, other work should be performed during this period.
Therefore, the work sequence should be adjusted to:
(1) Fetch onions for pot at (1,2), 12 steps
(2) Fetch onions for pot at (1,3), 18 steps
(3) Deliver onion soup for pot (1,2), 10 steps

Now, the rough work contents are:
(1) Fetch onions for pot at (1,2), 12 steps
(2) Deliver onion soup for pot (1,2), 10 steps
(3) Fetch onions for pot at (1,3), 8 steps
(4) Fetch onions for pot at (1,4), 10 steps
Please give me your answer as the example.

**We also provide an example of re-plan due to human feedback below**.

==== First round ====
Human: Please join me in making onion soup.
AI:
[Output of LLM1]
Cooking objectives: onion soup
AI's jobs:
Fetching vegetables: All pots.
Delivering food: All pots.

[Output of LLM2]
The pot in the scene is pot (3,6).
Based on the job description of AI in the key information, the rough work content for AI would be:
(1) Fetch onions for pot at (3,6)
(2) Deliver onion soup for pot (3,6)
Since there is only one pot in the scene, the rough tasks that the human player needs to complete would be: None.
...
[Ouptut of LLM5]
The work content and execution sequence of AI:
(1) Fetch onions for pot (3,6): Take the onion from position (5,5) and place it in the pot (3,6).
(2) Deliver onion soup for pot (3,6): Take the plate from (1, 4), then take the food from the pot (3, 6), and finally deliver it to the delivery port (3, 1).
The work content and execution sequence of Human:
None

[Human feedback]
You are only responsible for putting the onion into the pot,

and do not take onions from the onion dots below.

==== Second round ====
...
AI:
[Ouput of Session 1]
Cooking objectives: onion soup

AI's jobs:
Picking up vegetables: All pots.
Delivering food: Not mentioned.
...
[Ouput of Session 5]
The work content and execution sequence of AI:
(1) Fetch onions for pot at (3,6): Take the onion from position (1,7) and place it in the pot (3,6).
The work content and execution sequence of Human:
(1) Deliver onion soup for pot (3,6): Take the plate from (1, 4), then take the food from the pot (3, 6), and finally deliver it to the delivery port (3, 1).

