# OpenReview forum: "Efficient Human-AI Coordination via Preparatory Language-based Convention"
_ICLR.cc/2024/Conference — Submitted to ICLR 2024_

### Official Review · Reviewer_fw9C · 2023-10-29

**Soundness:** 2 fair
**Presentation:** 2 fair
**Contribution:** 2 fair
**Rating:** 5
**Confidence:** 4

**Summary:**

The paper considers the use of LLMs to facilitate human-AI coordination more intuitively. Specifically, the paper considers the possibility of using the LLMs as a way to set up a convention of how the task will be carried out. In the context of the paper, the convention takes the form of a task-level plan which lays out the different roles and actions to be carried out by each participant. To allow their method to support more complex scenarios, they use an approach to decompose the overall problem into sub-problems and then solve each one sequentially. The method is evaluated both using human proxies and with real human participants. The method is compared against multiple learning-based baselines.  Additionally, they performed an ablation study on their approach, showed the effectiveness of their method on other reasoning benchmarks, and showed how their method can achieve higher levels of human-AI value alignment.

**Strengths:**

I think the paper tackles an important problem, namely coordination in the MARL setting. The paper proposes an interesting approach to using LLMs in this context, i.e., coming up with a coordination plan. I also appreciate the fact that the authors took the trouble to run actual user studies, which are very important in validating such systems. Finally, I also like the fact that the authors acknowledge potential limitations the LLM-based reasoning systems might have with respect to reasoning and propose a method to address it.

**Weaknesses:**

In terms of the weaknesses, I had issues with clarity formulation – multiple terms were loosely used and not clearly formalized. There is some important information missing about the user studies. Next, there seem to be certain assumptions being made about the work whose implications are never clearly spelled out, and finally, I had some questions about the comparison between the current methods and the RL baselines. Below, I have provided a more detailed discussion of the specific concerns I had.

1. Clarity Issues: First off, there are multiple terms and concepts being referred to or introduced in the paper that are never fully formalized. Two specific ones that confused me were user preference and sequencing strategy. The paper talks about inputting human preferences in multiple places, and some instances of it are included in the example interactions. However, at no point is it made clear what exactly is human-preferences with respect to the underlying game formulations. Are these constraints over policies/total utility received by the user? Is this information already contained in the human reward function but just made more explicit in the input to the LLM? Secondly, the paper talks about problem decomposition and sequencing and how the one used in the paper is a general form that can be used in any task (Limitation section). But apart from some examples given in the figures, there is no exact formalization of this process. How did the authors prove that this decomposition is applicable to all possible extended-form Markov games? What happens when there is concurrency required between different tasks? Can each task be refined in isolation? How does stochasticity affect this process, and how do you account for task failure?

2. User Studies - While I really think it is great that the authors chose to do user studies, I have some concerns about the current user study. First off, I don’t see any mention of the user study being approved or granted exemption by an institutional review board. Similarly, there is no mention of the demographics and background of the participants, how they were recruited, and what the compensation was. Also, since there were only 12 participants and 16 conditions (method*setting), how were the studies counterbalanced? Finally, in the LLM setting, creation and validation of convention introduce new steps. Were any subjective evaluations performed on how much they liked the system? Also, the authors should have measured the cognitive load placed on the participants.

3. Assumptions: Two assumptions that are central to the success of the task are the ability to find high-level actions that can be executed by the low-level controller robustly and also make sense to the user. Here, both components are equally important because a high-level plan is shown to the user that is specified in terms of these high-level actions. If the human cannot make sense of or exactly predict how the agent will carry out these actions, the entire convention process will fall apart. While systems like SayCan make the former assumption (high-level skills), I believe the latter assumption is not crucial to the success of that system. Additionally, the method also expects access to a high-level abstraction of human actions (which is symmetric in Overcooked but may not be so in the most general case). Finally, there seems to be an additional assumption that even though the actions are abstract, any refinement of high-level conventions to low-level actions will be free of conflicts. This is usually hard to guarantee for more general coordination settings.

4. RL Action Space: Finally, it seems that the RL agent is learning policies and directly acting in the low-level action space. How does this allow for a fair comparison? Especially since the skills were learned from human demonstration, wouldn’t it naturally align with how humans would act? Also, did the same 12 participants provide the data from which the skills were learned?

**Questions:**

Please respond to the questions mentioned as part of the four points mentioned in the earlier section.

**Details Of Ethics Concerns:**

There is no mention of an IRB approval for the study discussed in the paper.

---

> ### Author Response · Authors · 2023-11-20
> **Sincerely thanks for your thoughtful comments! Response to Reviewer fw9C (1/2)**
>
> We sincerely thanks for you thoughtful comments. Below we have provided some clarifications to your main concerns. Further questions are always welcome!
>
> **Concern 1 Clarity Issues: Unclear usage of items and concepts, especially regarding user preference and sequencing strategy.**
>
> > **C1-1  Unclear about what exactly is human-preference**
>
> **A:** We sincerely apologize for any confusion caused by our unclear statements. Actually, the human-preference here means that the humans can have certain preference priorities regarding how to collaboratively accomplish the task and what specific roles they undertake. The same concept has also been adopted in existing works, such as [1, 2].
>
> More specifically, on the Overcooked, human-preference can be that the human prefers to utilize the left pot instead of the right pot, or the human prefers to make onion soup instead of tomato soup. We have added explanations about this concept in our revised version.
>
> > **C1-2 The paper talks about problem decomposition … is applicable to all possible extended-form Markov games?**
>
> **A:** We apologize for the unclear statement. In original statement, we intended to explain that our approach to task decomposition is relatively abstract and not strongly coupled to the Overcooked task, instead of claiming that it is applicable to all possible extended-form Markov games. We have modified our statement and incorporated it in Appendix F.
>
> > **C1-3 What happens when there is concurrency required between different tasks? Can each task be refined in isolation?**
>
> **A:** Although the Overcooked environment does not involve concurrency between different tasks, for new environment with this requirement, it is entirely feasible for HAPLAN to support this functionality. Actually, we only need to design the prompts to make the derived convention express between which tasks there exist concurrency.
>
> > **C1-4 How does stochasticity affect this process?**
>
> **A:** The planning of LLMs disregards stochasticity as the resulting high-level conventions are insensitive to it.  Instead, we expect the low-level skill policies can be robust to the stochasticity.
>
> On Overcooked environment, there does not exist stochasticity in the transition and reward functions. However, there can exist stochasticity in the agent’s observation because of the uncertainty in the partner’s state. We address this issue by collecting the demonstration segments with random-policy partner. How to train low-level skill policies that are robust to the stochasticity of the environment itself is an important future research direction.
>
> > **C1-5 How do you account for task failure?**
>
> **A:** When encountered with task failure, we would inspect the high-level convention and low-level skill policies respectively. Firstly, we can check whether there exist issues in the high-level convention. If so, we will regenerate the convention through providing human feedback. Similar adaptive replanning methods can be referenced in [3, 4]. If not, it indicates that there is something wrong with the low-level skill policies, and we will re-train the low-level skills to address this issue.
>
> **Concern 2 User Studies: Concerns about the user study**
>
> **A:** We sincerely appreciate this detailed feedback. We have provided clarification in our **Common Response to Q1**.
>
> **Concern 3 Assumptions: Some assumptions should be stated clearly**
>
> **A:** Thanks to your comment.
>
> - **(Two assumptions are central to the success of the task)** Thanks for this kind reminder. We acknowledge that these two assumptions are significant for HAPLAN and we have added this discussion in our revision. As mentioned by the reviewer, systems like SayCan [5] also requires the first assumption, which is kindly feasible in many scenarios. For the second assumption, despite the skill actions in SayCan are not conveyed to humans, they are also represented in the form of natural language, which implicitly indicates that humans are likely to be able to understand these skills.
> - **(Access to a high-level abstraction of human actions)** The high-level abstraction of human actions is related to the task itself. Since we focus on human-AI coordination tasks, we assume that we can have access to this portion of information and provide it in the examples preceding the prompt. We have stressed this assumption in our revision.
> - **(Refinement of high-level conventions will be free of conflicts)** We appreciate your thoughtful comment. However, we actually do not require this assumption. It is possible that conflicts arise during the refinement of high-level conventions. When such situation occurs, we consider these conventions as undesirable and expect to regenerate them through human feedback. For complex scenarios beyond immediate human recognition, humans can also avoid similar situations after an episode of failure.

---

> ### Author Response · Authors · 2023-11-20
> **Response to Reviewer fw9C (2/2)**
>
> **Concern 4 RL Action Space: Unfair comparison with RL agent**
>
> **A:** Sorry for our unclear statements. To make sure that the comparison with RL agents is fair, the data utilized in our experiments to learn low-level skills are not complete task trajectories; rather, it only consists of trajectory segments that accomplish specific skills, such as delivering onions to the pot in the top left corner. Thus, these demonstration data is purely functional and does not reflect any human behavior preference. Besides, these trajectory segments are actually collected by our authors, and not provided to the participants.
>
> **References**
>
> [1] Hu, Hengyuan, and Dorsa Sadigh. Language instructed reinforcement learning for human-ai coordination[J]. arXiv preprint arXiv:2304.07297, 2023.
>
> [2] Wang, Xihuai, et al. Quantifying Zero-shot Coordination Capability with Behavior Preferring Partners[J]. arXiv preprint arXiv:2310.05208, 2023.
>
> [3] Zhou, Siyuan, et al. Adaptive Online Replanning with Diffusion Models[C]. NeurIPS, 2023.
>
> [4] Guo, Yanjiang, et al. DoReMi: Grounding language model by detecting and recovering from plan-execution misalignment[J]. arXiv preprint arXiv:2307.00329, 2023.
>
> [5] Ahn, Michael, et al. Do as i can, not as i say: Grounding language in robotic affordances[J]. arXiv preprint arXiv:2204.01691, 2022.

---

> ### Author Response · Authors · 2023-11-22
> **Dear Reviewer fw9C, have our responses addressed your questions?**
>
> Dear Reviewer fw9C:
>
> We thank you again for your comments and hope our responses could address your questions. As the response system will end soon, please let us know if we missed anything. More questions on our paper are always welcomed. If there are no more questions, we will appreciate it if you can kindly raise the score.
>
> Sincerely yours,
>
> Authors of Paper3413

---

> > ### Comment · Reviewer_fw9C · 2023-11-22
> > **Response to reviewers**
> >
> > I really appreciate the detailed response. Correct me if I am wrong, but it seems the decomposition method is pretty ad hoc. The questions like concurrency, task failures, and stochasticity, were aimed at testing how the current decomposition method would handle such issues. From your response, it mostly seems that the decomposition method would need to be updated to account for such cases.

---

> ### Author Response · Authors · 2023-11-22
> **We sincerely appreciate your valuable feedback!**
>
> We are really delighted to see your new feedback! Firstly, we would like to clarify that the purpose of our work is NOT to provide a general task decomposition approach for various problems. Our main contribution is to propose a new paradigm for human-AI coordination through the integration of LLMs, as also pointed out by Reviewer nsKW. (Sorry for not highlighting it to you earlier)
>
> While the current decomposition method applies to all layouts on Overcooked, it is true that we would need to modify it when facing some new cases, like cases with concurrency requirement or stochasticity. However, we do not see this as a fundamental weakness but as an opportunity to enhance task coordination performance by incorporating our task priors. This capability is a direct result of introducing LLMs, which is clearly not allowed in the traditional Zero-Shot Coordination methods [1, 2, 3].
>
> Once again, we extend our gratitude for your feedback.
>
> **References**
>
> [1] Johannes Heinrich, Marc Lanctot, and David Silver. Fictitious self-play in extensive-form games. ICML, 2015.
>
> [2] Rui Zhao, Jinming Song, Yufeng Yuan, Haifeng Hu, Yang Gao, Yi Wu, Zhongqian Sun, and Wei
> Yang. Maximum entropy population-based training for zero-shot human-ai coordination. AAAI, 2023.
>
> [3] Chao Yu, Jiaxuan Gao, Weilin Liu, Botian Xu, Hao Tang, Jiaqi Yang, Yu Wang, and Yi Wu. Learning zero-shot cooperation with humans, assuming humans are biased. ICLR, 2023.

---

### Official Review · Reviewer_i2dJ · 2023-11-02

**Soundness:** 3 good
**Presentation:** 2 fair
**Contribution:** 3 good
**Rating:** 6
**Confidence:** 3

**Summary:**

The authors propose the use of large language models (LLMs) to facilitate coordination between humans and AI within human-AI teams through language-based conventions. Their findings indicate that LLMs are adept at aligning with humans who exhibit various coordination patterns, outperforming other baseline methods.

**Strengths:**

- The paper is well written with the exception of two subsections (5.3 and 5.4).
- Utilizing LLMs to come up with coordination strategies taking human preferences into account is an interesting direction.
- Experiments on the ability of the approach to coordinate with humans of different coordination patterns have been done using both real humans and proxy human models.

**Weaknesses:**

- The requirement for humans to decompose problems is a significant precondition. Additionally, humans are tasked with evaluating the LLM-generated plans and executing the tasks, which could be problematic in complex domains.
- The findings regarding the use of multiple sessions are somewhat expected, given that the problem decomposition is already done, significantly reducing the difficulty of the problem for the LLM.
- The details regarding the additional benchmark results are vague, particularly whether they refer to the coordination aspect or the use of multiple sessions.
- Depending on human evaluations to assess plans may be unreliable in complex domains, even when the evaluator is an expert.

**Questions:**

- Were there any cases where the LLM provided actions that were not at all executable in the environment? If so, how were they dealt?
- Figure 2 is hard to comprehend. What are the boxes in blue supposed to convey?
- Do the additional benchmark results have human-AI coordination as well? How are the respective tasks being divided between humans and the AI? It is unclear from the write-up.

---

> ### Author Response · Authors · 2023-11-20
> **Truly appreciate your valuable feedback! Below we have provided some clarifications in hopes that they can be helpful.**
>
> ### **Weaknesses**
>
> **W1 Unclear writing of Section 5.3 and 5.4**
>
> **A:** We apologize for the unclear writing of two subsections of 5.3 and 5.4. Actually, the experiments in these two subsections are both aimed to validate the effectiveness of utilizing multiple LLM sessions. Detailed clarification is provided in the response to **Q3**. We have improved the writing in our revision.
>
> **W2 Concern about the requirement for human to decompose problems and to evaluate & execute the plans generated by LLM**
>
> **A:** We acknowledge that it is a practical problem. Currently, we follow the common mechanisms using LLMs like in [1, 2], which have a reliance on human manual work. In the future, we will explore utilizing AI to decompose problems automatically and help evaluate the generated plans. Recently, there have been some efforts [3, 4] to explore related problems.
>
> **W3 The usage of multiple sessions are expected**
>
> **A:** Firstly, we sincerely appreciate the reviewer’s acknowledgement of the feasibility of utilizing multiple LLM sessions. Although this finding is somewhat expected, we have not come across any statements regarding this conclusion in the relevant literature. To our knowledge, we are the first to formally propose this finding. We hope it can inspire more research in the future.
>
> **W4 Vague details regarding the additional benchmark results**
>
> **A:** Apologize for the vague descriptions in this part. Actually, the additional benchmark results have nothing to do with the coordination aspect but to validate that utilizing multiple sessions can also help enhance the planning capability of LLMs in the reasoning tasks. We have made it clearer in our revision.
>
> **W5 Human evaluation may be unreliable**
>
> **A:** We acknowledge that this may be a limitation of our approach at this stage. As stated in our response to **W2**, in the future we will explore to utilize AI to help evaluate the plans in complex domains, as [3, 4] did.
>
> **References**
>
> [1]  Zhou D, Schärli N, Hou L, et al. Least-to-most prompting enables complex reasoning in large language models[J]. arXiv preprint arXiv:2205.10625, 2022.
>
> [2] Yang, Xinyi, et al. Human-in-the-loop Machine Translation with Large Language Model[J]. arXiv preprint arXiv:2310.08908, 2023.
>
> [3] Bai, Yuntao, et al. Constitutional ai: Harmlessness from ai feedback[J]. arXiv preprint arXiv:2212.08073, 2022.
>
> [4] Lee, Harrison, et al. RLAIF: Scaling reinforcement learning from human feedback with ai feedback[J]. arXiv preprint arXiv:2309.00267*,* 2023.
>
> ### **Questions**
>
> **Q1 Any cases that the LLM provided non-executable high-level actions?**
>
> **A:** Yes, there exist this kind of cases. Such as in the case shown in Figure 5(b), the initial plan generated by LLMs instructs the human player to fetch the onions from the bottom left corner, but this is not achievable due to the obstacle in the middle. When encountered this situation, the human can provide further feedback to help LLMs modify the plans.
>
> **Q2 Figure 2 is hard to comprehend. What are the boxes in blue supposed to convey?**
>
> **A:** We are sorry for any confusion in Figure 2. The boxes in blue means the questions that the LLM is expected to answer. We appreciate your feedback and we have provided a clearer depiction in the revised version.
>
> **Q3 Do the additional benchmark results have human-AI coordination as well? How are the respective tasks being divided between humans and the AI?**
>
> **A:**  Actually, the additional benchmark results are unrelated to the human-AI coordination, thus without need to divide the tasks between humans and AI. We apologize for this unclear statement again, and we have made modifications in our revised manuscript.

---

> ### Author Response · Authors · 2023-11-22
> **Dear Reviewer i2dJ, have our responses addressed your questions?**
>
> Dear Reviewer i2dJ:
>
> We thank you again for your comments and hope our responses could address your questions. As the response system will end soon, please let us know if we missed anything. More questions on our paper are always welcomed.
>
> Sincerely yours,
>
> Authors of Paper3413

---

### Official Review · Reviewer_nsKW · 2023-11-02

**Soundness:** 3 good
**Presentation:** 2 fair
**Contribution:** 3 good
**Rating:** 6
**Confidence:** 4

**Summary:**

This paper presents HAPLAN, and LLM-based approach to planning for ad hoc human-AI cooperation.  At a high-level, HAPLAN passes a textual description of the task (various scenarios in the Overcooked! environment in this case) to an LLM (ChatGPT), requesting that it return a detailed plan with instructions for both the human and the AI.  The AI's component of the plan is executed by a set of pre-trained, goal-conditioned policies for the various sub-tasks, specifically "fetching" and "delivering" items around the kitchen.  The human observes the plan before performing the task, and has the opportunity to provide corrective, natural language feedback to correct defects in the original plan.  Their experiments show that HAPLAN achieves significantly higher scores with real human partners and scripted AI partners than pervious methods proposed for ad hoc cooperation in Overcooked!.

The authors argue that one of the key contributions of the work is the idea of processing different stages of the planning process (high level planning, timing calculations, subtask ordering) in separate GPT sessions, which helps overcome the difficulties in reasoning about long conversation histories.  They support this argument with experiments (unrelated to ad hoc cooperation) on a set of benchmark reasoning tasks.

**Strengths:**

The main strength of the paper in my opinion is that it proposes (and experimentally validates) a new paradigm for practical human-AI cooperation that leverages the strengths of modern large language models to enable natural language interaction as part of the coordination process itself.  I could imagine that with further development, this approach could be useful for applications such as customer-support chatbots, for tasks requiring close coordination between the human and the chatbot to resolve a technical issue.

**Weaknesses:**

My main concern with the work is that the improved performance of HAPLAN relative to the baseline algorithms may have much less to do with its ability to cooperate with a variety of partners, and more to do with the superiority of the LLM-based hierarchical planning approach over the "flat" and "uninformed" RL algorithms used by the baseline approaches.

HAPLAN incorporates a great deal of human task knowledge, some of it through the natural language interface itself, but also through the use of pre-trained low-level policies that implicitly describe a decomposition of the high-level task into considerably simpler sub-tasks.  As Overcooked! can be challenging even without the ad hoc component (due to sparse rewards and the complexity of the policie needed to achieve them), it seems likely that the additional information available to HAPLAN would make a substantial difference in task performance against any partner.

To test this alternative explanation for the results, it would be useful to see how well HAPLAN compares to joint policies trained together for the overcooked task (such that each policy is a best-response to its partner).  It would also be helpful to provide more information about the "diversity" of strategies observed during human subjects experiments.

A related issue is that some important details of the experimental setup have been omitted.  Most significantly, it is unclear how plans are "stepped" during the interaction, that is, how the agent implementing the HAPLAN plan knows when an item has been fetched and delivered, and decides to move on to the next step?

Finally, while not a weakness of the HAPLAN approach itself, it is important to clarify for the reader that HAPLAN operates in a very different cooperation paradigm than the baselines.  While this paradigm, with a detailed conversational coordination phase prior to any physical interaction, may be suitable in some settings, it may not be useful in others (such as real time human-robot shared autonomy).

**Questions:**

1. How was the textual plan generated by HAPLAN "stepped" during execution?
2. Was there any real-time synchronization between the human and the AI?  For example, would the AI wait for the human if they were delayed in completing a prerequisite task?
3. How long were individual episodes of interaction (how many time steps, were the agents allowed to complete multiple dishes?)
4. For the scripted agents (Table 1), what information about the specific scripted policy was provided via the prompts?
5. Were humans allowed to construct their own initial prompts?  Rather than providing the instruction given to the human, could the human provide their own description of the role they planned to take?
6. How closely did human's observed behavior match the joint strategy they finally agreed upon?
7. It wasn't immediately clear, but were humans allowed to provide text-based feedback between episodes, or only in advance of episodes?
8. How much variance was there in the types of feedback humans provided?
9. How would a plan generated by HAPLAN compare to a joint policy trained via a cooperative MARL method?
10. How much prompt engineering was required here?  The prompts themselves are quite complex; are any results available with elements of these prompts removed?

---

> ### Author Response · Authors · 2023-11-20
> **Thanks for your interest in our work and valuable comments! Response to Reviewer nsKW (1/3)**
>
> ### **Weaknesses**
>
> **W1 Analysis on performance improvement of HAPLAN**
>
> **A:** Due to the strong hierarchical planning capabilities of LLMs and our proposed establishing conventions via feedback, HAPLAN can coordinate with diverse partners. To further analyze the performance improvement of HAPLAN, we additionally conducted two experiments:
>
> - **Performance of jointly learned policy**: We train a joint policy on Overcooked via MAPPO [1], the performance of which we denote as “Oracle” since both sides are best responses to each other. The results are listed in the following table. We can see that HAPLAN can achieve 75%~89% of the oracle performance thanks to planning by LLMs and human feedback. Although it is currently slightly less than 90%, HAPLAN still outperforms other baselines significantly. We hypothesize a significant factor affecting the overall performance is that the human participants are new to Overcooked task. We will conduct further research in the future to enhance performance.
>
>
>   |                       | Counter Circle   | Asymmetric Advantages | Soup Coordination | Distant Tomato   | Many Orders      |
>   | --------------------- | ---------------- | --------------------- | ----------------- | ---------------- | ---------------- |
>   | HAPLAN (First Round)  | 135.00$\pm$8.66  | 345.00$\pm$16.58      | 195.00$\pm$8.66   | 325.00$\pm$29.58 | 350.00$\pm$53.85 |
>   | HAPLAN (Second Round) | 160.00$\pm$14.14 | 360.00$\pm$24.49      | 205.00$\pm$8.66   | 355.00$\pm$16.58 | 380.00$\pm$50.99 |
>   | HAPLAN (Real Human)   | 170.00$\pm$17.32 | 385.00$\pm$21.79      | 215.00$\pm$16.58  | 370.00$\pm$22.36 | 410.00$\pm$51.96 |
>   | Oracle                | 225.19$\pm$18.31 | 447.63$\pm$8.11       | 240.67$\pm$4.13   | 481.51$\pm$54.08 | 462.76$\pm$12.71 |
>
> - **``diversity'' of strategies:**  We also visualize the human behaviors from human subject study. The result is shown in Figure 10 of our revised manuscript, where we can see that the participants encompass enough diversity. Considering that HAPLAN achieves the best performance in experiments, we conclude that our method can coordinate with diverse teammates.
>
> **W2 Constraints under certain problem setting**
>
> **A:** We greatly appreciate your suggestion and have made corresponding clarifications in the newest revision.
>
> ### **Questions**
>
> **Q1 How was the textual plan generated by HAPLAN ``stepped'' during execution?**
>
> **A:** The textual plan generated by HAPLAN is a sequence of skills for both human and AI. Humans can act seamlessly based on their natural language understanding of the plan. For AI, the execution requires to implement two additional functions, namely, **index function** and **completion function**. Here, the index function determines which low-level skill policy to call according to the text, while the completion function determines whether the current skill has been completed.
>
> We implement both the index function and complete function as rules in Overcooked, with the details provided in the supplementary materials. Investigating learning based implementation is an interesting direction, and we plan to leave it for future work.
>
> **Q2 Was there any real-time synchronization between the human and the AI?**
>
> **A:** Yes. Overcooked is a real-time synchronous environment, which means that the feedback ($s', r)$ for the next time step is provided only when both human and AI’s actions are determined.
>
> **Q3 How long were individual episodes of interaction?**
>
> **A:** The horizon of the individual episodes of interaction is 400 timesteps, and the agents are allowed to complete multiple dishes.

---

> ### Author Response · Authors · 2023-11-20
> **Response to Reviewer nsKW (2/3)**
>
> **Q4 For the scripted agents (Table 1), what information about the specific scripted policy was provided via the prompts?**
>
> **A:**   The prompts will convey the behavioral preference of each script agent to the LLMs. For example, for `Onion Placement` scripted agent, we will mention ``I will continuously place onions or tomatoes into the pot'' in the prompt.
>
> **Q5 Were humans allowed to construct their own initial prompts? Rather than providing the instruction given to the human, could the human provide their own description of the role they planned to take?**
>
> **A:** Yes, humans are allowed to construct their own initial prompts and provide their own description of the role they planned to take, all of which will be part of the prompts to the LLMs.
>
> **Q6 How closely did human’s observed behavior match the joint strategy they finally agreed upon?**
>
> **A:** Since both human and AI strictly follow the convention to take their actions, the human’s observed behavior fully match the convention (or the so-called joint strategy here) they finally agreed upon. The supplementary materials also provide a video demo of human-AI coordination.
>
> **Q7 It wasn’t immediately clear, but were humans allowed to provide text-based feedback between episodes, or only in advance of episodes?**
>
> **A:** Text-based feedback between episodes is allowed. Actually, in our real human study, we had each human participant cooperate with HAPLAN for three episodes, requiring human feedback at the beginning of each episode. To help humans focus more on the task, we currently do not allow them to provide feedback in an episode.
>
> **Q8 How much variance was there in the types of feedback humans provided?**
>
> **A:** Actually, the human feedbacks can reflect quite different human preferences. Taking the layout of Many Orders as an example, we can list the following different types of human preferences:
>
> | Human Preferences |
> | ------------------------------------------------------------ |
> | Responsible for placing ingredients in the left pot, not in charge of delivering the dishes. |
> | Responsible for placing ingredients in the right two pots, not in charge of delivering the dishes. |
> | Responsible for placing ingredients in all these three pots, not in charge of delivering the dishes. |
> | Responsible for delivering the dishes from the middle pot, not in charge of placing ingredients in the pots. |
> | Responsible for delivering the dishes from the left pot and right pot, not in charge of placing ingredients in the pots. |
> | Responsible for delivering the dishes from the three pots, not in charge of placing ingredients in the pots. |
> | Responsible for placing ingredients in the left pot and delivering the soup from the left pot. |
> | Responsible for placing ingredients in the left pot and delivering the soup from the left and middle pot. |
> | Responsible for placing ingredients in the middle and right pot and delivering the soup from the right pot. |
> | Responsible for placing ingredients in the right pot and delivering the soup from the three pots. |
>
> Taking the first type of human preference as an example, the human feedback can take various forms. We have compiled the occurrences observed in our experiments, which are as follows:
>
> | Human Feedback |
> | ------------------------------------------------------------ |
> | Please use the two pots on the right to make tomato soup and, by the way, take care of delivering the vegetables to the pot on the left. |
> | Use the two pots on the right to make tomato soup and also be responsible for delivering vegetables to the pot on the left. |
> | Cook tomato soup with the two pots on the right and, at the same time, be in charge of delivering vegetables to the pot on the left. |
> | Make tomato soup with the two pots on the right, and make sure to handle the vegetable delivery for the pot on the left as well. |
> | The two pots on the right are for making tomato soup, and you also need to take care of delivering vegetables to the pot on the left. |
> | Please use the two pots on the right to cook tomato soup and, in the meantime, take care of delivering vegetables to the pot on the left. |
> | Cook tomato soup with the two pots on the right, and also be responsible for delivering vegetables to the pot on the left. |
> | Please use the two pots on the right to make tomato soup, and also be responsible for delivering vegetables to the pot on the left. |
> | Simmer tomato soup with the two pots on the right, and also complete the vegetable delivery for the pot on the left. |
> | Please use the two pots on the right to make tomato soup and take care of the vegetable delivery task for the pot on the left. |

---

> ### Author Response · Authors · 2023-11-20
> **Response to Reviewer nsKW (3/3)**
>
> **Q9 How would a plan generated by HAPLAN compare to a joint policy trained via a cooperative MARL method?**
>
> **A:** Overall, HAPLAN can attain a performance ranging from 75% to 89% compared to a jointly trained policy using cooperative MARL methods. More details are provided in our response to **W1**.
>
> **Q10 How much prompt engineering was required here? The prompts themselves are quite complex; are any results available with elements of these prompts removed?**
>
> **A:** We thanks for your valuable feedback. However, we believe we have already used the minimal possible version of prompts and it is hard to further reduce them.
>
> Considering the complexity of the problem itself, we decompose the whole problem into several sub-problems and design specific prompts for a separate LLM session to solve each sub-problem. This kind of task decomposition has been demonstrated effective to improve the reasoning quality of the LLMs in Least-to-Most [2]. The prompts for each session contributes to the overall completion of the entire task and are indispensable.
>
> Besides, it is also inappropriate to remove any element in the prompts for one specific session. Taking the prompts for Session 1 in our Appendix G as an example, the first part `In a collaborative cooking game, ... the pot is onion` introduces the task background knowledge to the LLM, the second part `For Example 1: The instructions ... the middle pot` provides some demonstrations for the subproblem, while the third part `Now, the instructions ... examples` proposes the concrete question to the LLM.
>
> Actually, part 1 and part 3 are obviously necessary for the LLM session to solve the subproblem, while part 2 as demonstrations have already been proved effective in OpenAI’s official document [https://platform.openai.com/docs/guides/prompt-engineering/strategy-write-clear-instructions](https://platform.openai.com/docs/guides/prompt-engineering/strategy-write-clear-instructions). Moreover, we claim that we have already made the prompts design as concise as possible while ensuring the functionality of these prompts, as redundancy is unnecessary.
>
> **References**
>
> [1] Yu C, Velu A, Vinitsky E, et al. The surprising effectiveness of ppo in cooperative multi-agent games[C]. NeurIPS, 2022.
>
> [2] Zhou D, Schärli N, Hou L, et al. Least-to-most prompting enables complex reasoning in large language models[J]. arXiv preprint arXiv:2205.10625, 2022.

---

> ### Author Response · Authors · 2023-11-22
> **Dear Reviewer nsKW, have our responses addressed your questions?**
>
> Dear Reviewer nsKW:
>
> We thank you again for your comments and hope our responses could address your questions. As the response system will end soon, please let us know if we missed anything. More questions on our paper are always welcomed.
>
> Sincerely yours,
>
> Authors of Paper3413

---

> > ### Comment · Reviewer_nsKW · 2023-11-22
> > **Response to rebuttals and other reviews**
> >
> > Thank you for taking the time to respond to my questions in detail.  I believe that you have addressed one of my concerns, which was that in your experiments the HAPLAN agent did not face a significant coordination challenge once a human-AI plan had been generated.  The apparent variety of human preferences indicates that there was in fact significant diversity in the strategies followed by different human partners.
> >
> > Looking at the other three reviews, however, I tend to agree with the assessment that the HAPLAN approach would require significant effort to generalise to other tasks beyond overcooked.  I believe there are two things that future work could do make a stronger case that HAPLAN-like agents could be a practical solution to real-world human-AI coordination problems:
> > 1. First, formalise HAPLAN in domain-independent terms, and make it clear what would be required to apply the same approach in a different environment.  It appears to me that this would require designing (or training) a new "plan execution module" that translates textual plans to low-level actions, as well as a new set of prompts to describe the new planning problem itself.  Little of the engineering work that has gone into HAPLAN could be directly transferred to a new task as is.  What remains is the abstract process of plan generation and interactive plan refinement, which could serve as a template for new agents in other domains.
> > 2. Second, implement a HAPLAN agent in a second domain, preferably one that is closer to "real world" conditions than the benchmark Overcooked domain.  The goal here would be to show that not only do the planning abilities of LLMs generalise, but that humans can (and are willing) to follow AI-generated plans in practical settings.
> >
> > At present , I view the work as a preliminary proof-of-concept of the use of LLMs to support ad hoc cooperation, and have adjusted my score to reflect this.

---

> > > ### Author Response · Authors · 2023-11-23
> > > **Sincerely thanks for your new feedback!**
> > >
> > > First of all, we are really glad to see your two acknowledgements.
> > >
> > > 1. The human-AI coordination challenge can be solved effectively by HAPLAN when a convention is obtained, which is exactly our main motivation.
> > > 2. HAPLAN is equipped with the ability to cooperate with diverse human partners, which we believe is a great inspiration for the human-AI coordination community.
> > >
> > > Besides, we sincerely appreciate your further comments, and we provide some clarifications below:
> > >
> > > > **Concern about the effort required to apply HAPLAN to other environments.**
> > >
> > > To begin with, we would like to express our gratitude for your agreement that our abstract process of plan generation and interactive plan refinement could serve as a template for other domains. This could to a large extent reduce the effort required to adapt to new domains. While we acknowledge the necessity of designing new plan execution modules and prompts for new domains, we do NOT consider it a fundamental issue:
> > >
> > > 1. It is true that we need to re-train the low-level skills to help translate the textual-plan into low level actions when in a new domain. However, the same requirement can be exactly observed in SayCan [1], which is published in CoRL 2023. As stated in Appendix C.2 in SayCan, it consumes 27 hours of 16 TPUv3 chips to train the skill policies using BC. We believe this cost is worth it, especially considering the performance improvement it has brought to SayCan. Similarly, in our work, it has helped HAPLAN achieve coordination performance superior to previous methods.
> > > 2. Currently, NONE of existing works on manipulating LLMs had given such general prompts, and the prompt engineering seems to be **indispensable** to manipulate the LLMs effectively. SayCan [1] requires designing new prompts and training new low-level skills when meeting new domains. ELLM [2] requires detailed prompt engineering to provide specific task context and describe the agent state, thus to guide agent learning. LLMTIME [3] requires to design prompts to describe the background information and task goal to conduct time forecasting for new problems. Moreover, as stated in our response, and agreed upon by Reviewer nmM1, the prompt engineering is not a big issue because it is becoming progressively easier with the advances of the community [4, 5, 6] and tutorial on the Internet, such as [7, 8, 9].
> > >
> > > > **Experiments on a second domain**
> > >
> > > We sincerely thank the reviewer for this kind suggestion. Due to the time constraint during this discussion, we plan to consider this as our future work.
> > >
> > > Actually, the reason that we chose Overcooked environment is that, currently the majority of human-AI coordination research [7, 8, 9] only considers empirical validations on this environment. We follow the same experimental setup to ensure a fair comparison with them. Also, our approach has successfully outperformed previous baseline algorithms in various layouts on Overcooked, which to a large extent, validates the effectiveness of our approach.
> > >
> > > Once again, we sincerely thanks for your feedback!
> > >
> > > **References**
> > >
> > > [1] Brohan A, Chebotar Y, Finn C, et al. Do as i can, not as i say: Grounding language in robotic affordances[C]. Conference on Robot Learning, 2023.
> > >
> > > [2] Du Y, Watkins O, Wang Z, et al. Guiding Pretraining in Reinforcement Learning with Large Language Models[C]. ICML, 2023.
> > >
> > > [3] Gruver N, Finzi M A, Qiu S, et al. Large Language Models Are Zero-Shot Time Series Forecasters[C]. NeurIPS, 2023.
> > >
> > > [4] Wei J, Wang X, Schuurmans D, et al. Chain-of-thought prompting elicits reasoning in large language models[J]. Advances in Neural Information Processing Systems, 2022, 35: 24824-24837.
> > >
> > > [5] Zhou D, Schärli N, Hou L, et al. Least-to-most prompting enables complex reasoning in large language models[J]. arXiv preprint arXiv:2205.10625, 2022.
> > >
> > > [6] Yao S, Yu D, Zhao J, et al. Tree of thoughts: Deliberate problem solving with large language models[J]. arXiv preprint arXiv:2305.10601, 2023.
> > >
> > > [7] OpenAI. Prompt engineering. https://platform.openai.com/docs/guides/prompt-engineering.
> > >
> > > [8] Isa Fulford, Andrew Ng. ChatGPT Prompt Engineering for Developers**.** https://learn.deeplearning.ai/chatgpt-prompt-eng/
> > >
> > > [9] Lilian Weng. Prompt Engineering. https://lilianweng.github.io/posts/2023-03-15-prompt-engineering/

---

### Official Review · Reviewer_nmM1 · 2023-11-03

**Soundness:** 2 fair
**Presentation:** 2 fair
**Contribution:** 2 fair
**Rating:** 5
**Confidence:** 3

**Summary:**

In this paper, the authors present an approach to have humans coordinate with large language models via giving instructions to the LLM prior to the interaction (referred to as establishing a convention in the paper). These instructions are then iterated on during a sequence of interactions. The LLM is controlled via complex prompts (given in the supplementary materials) for each task which are translated into in-game actions via developer-specified low-level skills learned through demonstrations. The results indicate that the authors' approach (HAPLAN) meets or slightly exceeds the performance of the baselines, with greater effects on later rounds.

**Strengths:**

There is certainly a need to explore better ways for human users to interact with and controller Transformer-based models. This is also, to the best of my knowledge, an original approach for the Overcooked task specifically. The quality in terms of the amount of engineering work and time cost for the human subject study is also impressive. The results are likely to be of interest to those working with the Overcooked tasks or those with the resources to replicate the work HAPLAN requires for each domain.

**Weaknesses:**

This paper has three major weaknesses.

First is the approach itself. This is not clearly conveyed in the text of the paper, certainly not in sufficient detail for replication. As far as I can tell, the process is that the authors come up with a domain-specific sequence of prompts for the sequence of prompts in a domain. This includes coming up with a set of low-level skills to refer to in these prompts. They then train these low-level skills via imitation learning. Then, during an interaction, they use an LLM to interact with the user and based on the user's prompts/conventions, the LLM produces an output plan referencing the low-level skills, which is then executed. If this is the case, this is an approach that has a lot of barriers to generalization, requiring human expertise and significant development time. It's also not guaranteed to generalize to every human-AI interaction domain, as it may not always be possible to break a problem into sequences or tasks or into low-level skills. The novelty of this approach is also fairly low, relying on putting together existing approaches.

Second is the human subject study methodology. There's no clarity in the paper in terms of what this methodology was. While the supplementary materials indicate that efforts were put forth to attempt to decrease bias, it's unclear what this means exactly. There are many possible sources of bias in terms of what population was recruited from, how they were compensated, what instructions they were given, and so on. Clarification on these points, ideally through a complete breakdown of the methodology, is necessary in order to avoid any potential that the results might be tainted by bias.

Third is the results. The improvement in terms of the results is fairly marginal for what appears to be a much more complex and engineering-intense approach. Further, almost all the baselines see improvement over the three rounds, so it's unclear to what extent the convention is helping. The inclusion of a version of HAPLAN without the convention might have helped clarify this.

**Questions:**

1. Am I correct in my understanding of HAPLAN?
2. Am I correct in my understanding of the development/design and engineering work needed to adapt HAPLAN to a new domain?
3. What was the methodology of the human subject study?
4. Is the improvement of HAPLAN significant?

**Details Of Ethics Concerns:**

My concerns have been addressed with additional details.

---

> ### Author Response · Authors · 2023-11-20
> **Thanks for your detailed comments! Response to Reviewer nmM1 (1/3)**
>
> ### **Weaknesses**
>
> **W1 Concerns about the development and engineering work needed to adapt HAPLAN to a new domain**
>
> **A:** Thanks for your insightful comment. We admit that it may be inevitable to design new prompts when facing a brand-new domain. However, we claim that it is not a fatal drawback of HAPLAN:
>
> - Firstly, we would like to re-claim that the purpose of our work is NOT to provide a general prompt suitable for every human-AI coordination task. Instead, just as pointed out by Reviewer nsKW, our main contribution is that we propose a new paradigm for human-AI coordination that leverages the strengths of LLMs.
> - Secondly, we agree that to guide LLMs to perform in a manner aligned with human expectations needs carefully designed prompts. However, with the publications of some recent works on prompting LLMs, such as Chain-of-Thought [1], Least-to-Most [2], Tree-of-Thought [3], etc. and tutorials on the Internet, such as [4,5,6], we can anticipate that prompt engineering is becoming progressively easier.
> - Moreover, we have provided the complete prompts in Appendix F used in our experiments, which we believe can serve as good references for new domains. Providing clear descriptions of task details and rules, incorporating human preferences, and leveraging techniques like Chain-of-Thought and using multiple sessions proposed in our work can all be helpful.
>
> **W2 Whether it is always possible to break a problem into sequences or tasks or into low-level skills**
>
> **A:** We appreciate your thoughtful comment. We assume that the task can be divided into sequences due to the sequential property of Dec-POMDPs. Moreover, we argue that the task can be divided into sub-tasks or low-level skills is not a strong assumption. Many problems could be divided into sub-tasks [7] or hierarchical structure [8], as the complex world is decomposable or nearly decomposable[9, 10]. Below are more evidences:
>
> - On one hand, most existing collaborative problems can be structurally decomposed, such as SMAC [11] which can be divided based on which specific target to attack; Google Research Football [12] which can be divided on multiple subtasks or skills, like interception, passing and shooting; not to mention the Overcooked we use.
> - On the other hand, this assumption is commonly used in existing studies. In SayCan [13], as pointed out by Reviewer fw9C, the authors assume that we can pre-train low-level reusable skills, like “pick up the apple”. Least-to-most [2] decomposes a hard problem into several easier subproblems to facilitate the reasoning of LLMs. Moreover, a large body of existing multi-agent hierarchical reinforcement learning methods, such as [14], dealing with coordination tasks also make similar assumptions.
>
> For some non-decomposable setting, we leave it as an significant future work.
>
> **W3 Novelty of the approach**
>
> **A:** Our **main contribution**, as mentioned by Reviewer nsKW, lies in proposing a novel paradigm for human-AI coordination. To our best knowledge, this is the first work that utilizes LLMs to enable natural language interaction as part of the coordination process itself.
>
> Through our proposed decomposing the problem, solving subproblems via multiple LLM sessions, and constructing conventions via human feedback, HAPLAN significantly enhances the efficiency of human-AI coordination.
>
> **W4 Clarification on the methodology of the human subject study?**
>
> **A:** We apologize for any confusion caused by our unclear statement. We have provided clarification to human subject study in **Common Response Q1**. Hope it will help.

---

> ### Author Response · Authors · 2023-11-20
> **Response to Reviewer nmM1 (2/3)**
>
> **W5 To what extent do the conventions help?**
>
> **A:** We appreciate your thoughtful comment. Our response is two-fold:
>
> - First, the baselines in our experiments do not utilize conventions, as they are unable to explicitly incorporate feedback. The performance improvement of them across three rounds is mainly because that humans become more familiar with the task and optimize their own decision-making in the game.
> - To further check **to what extent** the convention is helping, we externally include one baseline denoted as HAPLAN w/o convention, which disables developing conventions via human feedback. The result is shown in the subsequent table, where we can see that HAPLAN w/o convention suffers from a significant performance decline especially at the final round, indicating the effectiveness of convention.
>
> |              |                       | Counter Circle | Asymmetric Advantages | Soup Coordination | Distant Tomato | Many Orders  |
> | ------------ | --------------------- | -------------- | --------------------- | ----------------- | -------------- | ------------ |
> | First Round  | HAPLAN w/o convention | 135.00$\pm$21.79 | 340.00$\pm$14.14        | 195.00$\pm$16.58    | 325.00$\pm$25.98   | 355.00$\pm$45.55 |
> |              | HAPLAN                | 135.00$\pm$8.66  | 345.00$\pm$16.58        | 195.00$\pm$8.66     | 325.00$\pm$29.58   | 350.00$\pm$53.85 |
> | Second Round | HAPLAN w/o convention | 145.00$\pm$16.58 | 350.00$\pm$10.00        | 205.00$\pm$16.58    | 335.00$\pm$16.58   | 365.00$\pm$29.58 |
> |              | HAPLAN                | 160.00$\pm$14.14 | 360.00$\pm$24.49        | 205.00$\pm$8.66     | 355.00$\pm$16.58   | 380.00$\pm$50.99 |
> | Third Round  | HAPLAN w/o convention | 150.00$\pm$22.36 | 355.00$\pm$8.66         | 205.00$\pm$8.66     | 340.00$\pm$28.28   | 370.00$\pm$33.16 |
> |              | HAPLAN                | 170.00$\pm$17.32 | 385.00$\pm$21.79        | 215.00$\pm$16.58    | 370.00$\pm$22.36   | 410.00$\pm$51.96 |
>
> **References**
>
> [1] Wei J, Wang X, Schuurmans D, et al. Chain-of-thought prompting elicits reasoning in large language models[J]. Advances in Neural Information Processing Systems, 2022, 35: 24824-24837.
>
> [2] Zhou D, Schärli N, Hou L, et al. Least-to-most prompting enables complex reasoning in large language models[J]. arXiv preprint arXiv:2205.10625, 2022.
>
> [3] Yao S, Yu D, Zhao J, et al. Tree of thoughts: Deliberate problem solving with large language models[J]. arXiv preprint arXiv:2305.10601, 2023.
>
> [4] OpenAI. Prompt engineering. [https://platform.openai.com/docs/guides/prompt-engineering](https://platform.openai.com/docs/guides/prompt-engineering).
>
> [5] Isa Fulford, Andrew Ng. ChatGPT Prompt Engineering for Developers**.** [https://learn.deeplearning.ai/chatgpt-prompt-eng/](https://learn.deeplearning.ai/chatgpt-prompt-eng/)
>
> [6] Lilian Weng. Prompt Engineering. [https://lilianweng.github.io/posts/2023-03-15-prompt-engineering/](https://lilianweng.github.io/posts/2023-03-15-prompt-engineering/)
>
> [7] Guestrin, Carlos, et al. Efficient solution algorithms for factored MDPs[J]. Journal of Artificial Intelligence Research*,* 2003.
>
> [8] Pateria, Shubham, et al. Hierarchical reinforcement learning: A comprehensive survey[J]. ACM Computing Surveys, 2021.
>
> [9] Zhang, Chongjie. Scaling multi-agent learning in complex environments. University of Massachusetts Amherst, 2011.
>
> [10] Yuan, Lei, et al. Multi-Agent Concentrative Coordination with Decentralized Task Representation[C]. IJCAI, 2022.
>
> [11] Samvelyan, Mikayel, et al. The starcraft multi-agent challenge[J]. arXiv preprint arXiv:1902.04043, 2019.
>
> [12] Kurach, Karol, et al. Google research football: A novel reinforcement learning environment[C]. AAAI, 2020.
>
> [13] Ahn, Michael, et al. Do as i can, not as i say: Grounding language in robotic affordances[J]. arXiv preprint arXiv:2204.01691, 2022.
>
> [14] Zhang F, Jia C, Li Y C, et al. Discovering generalizable multi-agent coordination skills from multi-task offline data[C], ICLR, 2022.

---

> ### Author Response · Authors · 2023-11-20
> **Response to Reviewer nmM1 (3/3)**
>
> ### **Questions**
>
> **Q1, 2 Am I correct in my understanding of HAPLAN? Am I correct in my understanding of the development/design and engineering work needed to adapt HAPLAN to a new domain?**
>
> **A:** Please refer to our reponse to W1.
>
> **Q3 What was the methodology of the human subject study?**
>
> **A:** Please refer to our common response to human subject study.
>
> **Q4 Is the improvement of HAPLAN significant?**
>
> **A:** To check whether the improvement of HAPLAN is significant, we compare the percentage of performance improvement of HAPLAN over the second-best baseline with both human proxy models and real humans. The results are presented in the following two tables.
>
> | (Human Proxy) Layout | Partner                     | Performance improvement to the second-best baseline |
> | -------------------- | --------------------------- | --------------------------------------------------- |
> | Soup Coordination    | Onion Placement & Delivery  | 10.03%                                              |
> | Soup Coordination    | Tomato Placement & Delivery | 14.61%                                              |
> | Distant Tomato       | Tomato Placement            | 41.18%                                              |
> | Distant Tomato       | Tomato Placement & Delivery | 26.82%                                              |
> | Many Orders          | Delivery                    | 15.89%                                              |
>
> | (Real Human) Layout                                  | Counter Circle | Asymmetric Advantages | Soup Coordination | Distant Tomato | Many Orders |
> | ---------------------------------------------------- | -------------- | --------------------- | ----------------- | -------------- | ----------- |
> | Performance improvement to the second-based baseline | 3.03%          | 4.05%                 | 7.50%             | 5.71%          | 9.33%       |
>
> From the above tables, we see that: When tested with human proxy models, HAPLAN achieved a performance improvement of **more than 10%** compared to the second-best baseline in **half of the scenarios**, and a significant performance improvement of **more than 40%** in Distant Tomato; when tested with real humans, HAPLAN surpassed all other baselines in all of the layouts.

---

> > ### Comment · Reviewer_nmM1 · 2023-11-21
> > **Re: Response to Reviewer nmM1**
> >
> > I thank the reviewers for their thoughtful and exhaustive response. In summary, I had identified three initial weaknesses. First was the approach. I appreciate the author's argument that prompt engineering will become easier and that most tasks can be broken into subtasks. In addition, while I am still concerned about the novelty of the approach in terms of explicitly setting a convention being a relatively minor variation of existing LLM agent interaction, the inclusion of a baseline does help to address this.
> >
> > Second, I had concerns with the lack of human subject study methodology description. I appreciate the author's explanation of the study methodology. However, I still retain two concerns. First, "we ensured similarities in age and gender distribution among these groups" makes it sound as though the researchers intentionally placed certain subjects in certain groups, which is a clear bias risk. Second, that the explanation seems to suggest that only 5 participants were in each condition, which is insufficient for statistical significance.
> >
> > Third was the concern around the results. I very much appreciate the inclusion of the additional baseline. An increase in 10% may not be significant in terms of an increase, particularly with such large std dev. If there are at least 10 participants in each condition a statistical test would be helpful here.
> >
> > Overall, all of my concerns have been at least partially addressed but I retain some questions. I'm upgrading my review to reflect this.

---

> > > ### Author Response · Authors · 2023-11-21
> > > **Sincerely thanks for your timely feedback**
> > >
> > > Sincerely thanks for your timely feedback and  kindly raising the score. We are glad that our response could be helpful, and below we provide some further clarifications to your remaining concerns.
> > >
> > > > **Concern 1 Any certain subjects intentionally placed in certain groups?**
> > >
> > > **A:** Thanks for proposing this concern. In fact, this sentence is meant that we tried to make the distribution of age and gender as similar as possible between different groups thus to reduce differences between them. These participants are all campus students with fairly similar backgrounds, further reducing the possible bias.
> > >
> > > Moreover, in each Overcooked layout, we will randomly assign these groups to different algorithms for testing, rather than binding each group to a specific algorithm throughout the entire experiment.
> > >
> > > > **Concern 2 Concern about the results due to relatively few real human participants**
> > >
> > > **A:** At current stage, due to recruitment and experimental cost, our real human experiments are indeed not conducted on a large scale. However, we have tried to make our conclusion more convincing by reusing these participants across more layouts (5 layouts), providing a broader set of experimental results. Overall, our method has shown superior results across all layouts.
> > >
> > > In the future, if conditions permit, we look forward to conducting experiments with more real human participants to further validate and enhance our approach.
> > >
> > > Finally, we thank you again for your feedback! Any further questions are always welcome.

---

### Author Response · Authors · 2023-11-20
**Common Response (2/2)**

- **Compensation Rules:** Basically, we provide 15 dollars for each participant for compensation. Meanwhile, to encourage participants to conduct testing seriously, we provide extra 10 dollars for the top three participants based on their total score achieved during the gameplay.

- **Coordinating with HAPLAN:** When conducting experiments with human participants using HAPLAN, we allow natural language interaction between human and AI. The detailed steps are as follows:

  - Firstly, the human can tell HAPLAN his/her preference about coordination, e.g., “I will make onion soup, and the AI will handle food delivery”. Then these words will be filled in the prompt and conveyed to the LLMs.

  - Then, the LLMs will generate a detailed plan using multiple sessions. This process involves solving multiple subproblems as depicted in Figure 2 in our manuscript. The final plan describes what human and AI are expected to do. Below is an example:

    > The work content and execution sequence of AI: (1) Fetch onions for pot at (3,6): Take the onion from position (1,7) and place it in the pot (3,6).

    > The work content and execution sequence of Human: (1) Deliver onion soup for pot (3,6): Take the plate from (1, 4), then take the food from the pot (3, 6), and finally deliver it to the delivery port (3, 1).

  - If a participant deems the plan unsatisfactory, he/she can offer feedback and request the LLMs to regenerate one.

  - If the plan is accepted, the participant can execute the plan itself, while the AI will execute the plan with low-level skills. How the AI execute the plan is described in our answer to **Q1 of Reviewer nsKW** in detail.

- **Bias Avoidance:**

  - To avoid bias introduced in real human study as much as possible, we recruit participants with similar background, all being campus students aged 19 to 22. These participants all possess basic comprehension and manual skills.
  - Besides, we divided participants into 5 groups and let each group cooperate with one AI method (total 5 AI methods including HAPLAN w/o convention). To avoid unfair comparison between different AI methods, we ensured similarities in age and gender distribution among these groups. Additionally, we randomly assigned each group to cooperate with one specific AI method in each experimental layout.

- **Experiment Setup:**

  - Considering that we included a baseline comparison of HAPLAN w/o convention in the rebuttal, we additionally recruited new participants during the rebuttal process. Combining the experimental results of the new participants with those from the previous ones, we have aggregated test results from a total of 20 participants. Specifically, these 20 participants were randomly divided into 5 groups. Across each layout, we assigned these 5 groups of participants randomly to different AI methods (total 5 AI methods including HAPLAN w/o convention) for testing.
  - Special note that we have increased the number of participants in each group (4 individuals per group now compared to 3 before) to further mitigate bias in the rebuttal process.

---

### Author Response · Authors · 2023-11-20
**Common Response (1/2)**

Thank all of you for your thoughtful and inspiring comments. Here is a summary of major updates made to the revision:

1. We provided additional experimental results in Appendix D.1, D.2 and D.3, which respectively conduct ablation study on convention, compare HAPLAN with oracle collaborative performance and investigate the strategy diversity during real human experiments.
2. We presented more details about real human experiments in Appendix E.
3. We have added a more detailed discussion about the limitations and future work in Appendix F, incorporating our original paragraph and the reviewers’ suggestions.
4. Since we have to recruit new participants in rebuttal process to evaluate the added baseline ``HAPLAN w/o convention'', we also increased the number of test participants for other algorithms to further mitigate possible biases. Currently, we totally recruited 20 human participants and test each AI method with 4. We have updated the real human experimental results in the revision.
5. We have expanded our manuscript by including further explanations to mitigate potential confusion. These additions encompass clarification on the concept of human-preference, elucidation of the blue box in Figure 2, and extra description of additional benchmark results.

Note the revision made is marked as “blue” in the revised paper.

We hope that our response and revision address your concerns and questions. We are happy to provide further clarification if you have any additional concerns or comments.

**Q1 Details of Real Human Study (Reviewer nmM1, fw9C)**

**A:** We apologize for the lack of clarity regarding the details of the real human study, which has been mentioned by Reviewer nmM1 and fw9C. We illustrate the details of real human study here and have added it in our revision.

- **Recruitment Details:** We recruited human participants from our campus community, specifically by posting a recruitment notice seeking individuals to participate in a Gaming AI test. To mitigate the impact of participants' prior experience, we selected participants who have not played the Overcooked game. The volunteers we recruited fall within the age range of 19 to 22 years old.
- **Experiment Instructions:** To provide participants with a rough background of the experiments, we provide them with experiment instructions as follows:
  - **Basic operations**
    - The Overcooked game naturally requires human-AI coordination to achieve a high score. In this game, you will cooperate with an AI agent as a team.
      - There are six actions available in the game: up ("W"), down ("S"), left ("A"), right ("D"), no-operation ("F"), and interact ("G"). Each action consumes one time step, and a complete game consists of 400 time steps.
      - The relevant objects are onions, tomatoes, plates, and soups.
      - When the game starts, you can move (up, down, left, and right) towards the onion and interact with it. Once you interact with the onion, you will acquire it and be carrying it.
      - When you are carrying an onion, you can move to a pot and interact. By doing so, you will place the onion you are carrying into the pot.
      - Place three onions or tomatoes in a pot and wait for 20 time steps, then the ingredients will transform into a pot of delicious soup.
      - Now move to a plate and interact with it to carry the plate. Take the plate and move towards the cooked soup, then interact with it. This action will scoop the soup into the plate.
      - The final step is to deliver the vegetable soup to the delivery port, and you will complete an order successfully.
      - Upon completing each order, you will receive a reward of 20 points.
  - **Experimental settings**
    - In our experiment, you will cooperate with one agent teammate for three rounds on each Overcooked layout. These agents may require interaction with you or not.
      - When cooperating with agent that requires interaction, it is necessary to establish a convention with it through the interactive window before each episode. For example, you can say, "I will make onion soup, and the AI will handle food delivery." Based on this convention, the system will generate a detailed plan for you and your AI teammate. You have the option to either accept the plan, request a plan reformulation, or even change your convention.
      - When working with other agents, you will directly cooperate with them and strive to achieve high scores.

---

### Author Response · Authors · 2023-11-21
**Request for Further Review Feedback**

Dear Reviewers,

We sincerely appreciate the time and effort you have invested in reviewing our paper. We have carefully considered all your feedback and have endeavored to address each concern in our submitted response. As the discussion period is nearing its conclusion, we kindly request any further feedback you may have. Your insights are crucial in enhancing the quality of our work.

If you find that our responses have adequately addressed your concerns, we would be grateful for your consideration in possibly raising the rating of our work. On the other hand, should you have any additional questions or lingering concerns, please do not hesitate to contact us. We are more than willing to provide further clarifications and answer any additional questions you might have.

Once again, we extend our deepest gratitude for your valuable contributions to the review process.

Best Regards,

The Authors

---

### Meta-Review · Area_Chair_X8ct · 2023-12-03

**Metareview:**

This paper proposed a novel paradigm, preparatory language-based conventions, for human-AI coordinations. The main idea is to emply LLM to develop conventions/action plans to guide both humans and AI before coordination. Experiments conducted on Overcooked platform with real human participants demonstrate the effectiveness of the approach.

Overall all reviewers commend on the novel idea and the execution of the proposed research! The main reservations during rebuttal and discussion afterwards come from 1) the generalizability concern, and 2) the relatively small-scale human-subject experiments. For 1), while the paper does not claim to develop general prompts, given the domain application Overcooked is more of a proof-of-concept platform, the contribution of the paper would be more convincing if there are stronger indications/evidence that the approach could be generalized to other applications, e.g., through the suggestions by nsKW.  For 2), reviewers overall commend on the effort of conducting human-subject experiments, however, the relatively small scale of the experiment also limit the result significance.

**Justification For Why Not Higher Score:**

This is a solid paper. The main concerns come from that the approach is heavily tailored to this particular proof-of-concept domain and the relatively small-scale of the experiments. With one of the perspectives being better addressed, this paper would pass the bar for acceptance.

**Justification For Why Not Lower Score:**

N/A

---

### Decision · Program_Chairs · 2024-01-16

Reject